# An agricultural digital twin for mandarins demonstrates the potential for individualized agriculture

**Steven Kim** [1] ✉ **& Seong Heo** [2] ✉

A digital twin is a digital representation that closely resembles or replicates a real world object by combining interdisciplinary knowledge and advanced technologies. Digital twins have been applied to various fields, including to the agricultural field. Given big data and systematic data management, digital twins can be used for predicting future outcomes. In this study, we endeavor to create an agricultural digital twin using mandarins as a model crop. We employ an Open API to aggregate data from various sources across Jeju Island, covering an area of approximately 185,000 hectares. The collected data are visualized and analyzed at regional, inter-orchard, and intra-orchard scales. We observe that the intra-orchard analysis explains the variation of fruit quality substantially more than the inter-orchard analysis. Our data visualization and analysis, incorporating statistical models and machine learning algorithms, demonstrate the potential use of agricultural digital twins in the future, particularly in the context of micro-precision and individualized agriculture. This concept extends the current management practices based on data-driven decisions, and it offers a glimpse into the future of individualized agriculture by enabling customized treatment for plants, akin to personalized medicine for humans.

Since the concept of a digital twin (DT) was emerged by Grieves M[1]., it has been introduced in various fields including aerospace, automotive, manufacturing, construction, real estate, health, medicine, and agriculture[2–6]. Though DT has broad meanings in various fields, it is generally defined as the implementation of virtual counterparts of real-world physical systems in a digital environment. It often allows users to simulate, model, and analyze data to make informed decisions[7]. DT relies on the integration of state-of-the-art technologies including information and communication technologies (ICT), Internet of things (IoT), remote sensing, geographic information systems (GIS), big data analytics, and artificial intelligence (AI)[8,9]. The ICT provides the infrastructure and communication networks necessary for the acquisition, aggregation, storage, and analysis of data from IoT devices and remote sensing, thereby farmers can access and use digital platforms for precision farming and crop management. Wireless IoT devices such as

sensors collect agricultural data including weather conditions, soil moisture, and crop physiological information[10]. Furthermore, digital imagery produced from UAV and satellites (remote sensing) has led to a paradigm shift for farmers and researchers from an approach of homogeneous management of heterogeneous fields to one of a heterogeneous management of heterogeneous fields (soil fertility, soil moisture, plant pathogens etc.)[11]. For this paradigm shift, the agricultural data should be managed in combination with longitudinal data and geospatial data for implementing agricultural practices at the right time and location. In particular, geospatial data enable farmers to apply input materials based on crop needs on a precise site-specific basis[12]. Given big data and systematic data management, AI such as machine learning and deep learning algorithms can be used for predictions and data-driven decisions. The results provide farmers with insights for improving decision making and supplying the required

[1]Department of Mathematics and Statistics, California State University, Monterey Bay, Seaside, CA 93955, USA. [2]Department of Horticulture, Kongju National University, Yesan 32439, Republic of Korea. ✉e-mail: stkim@csumb.edu; heoseong@kongju.ac.kr

input resources (water, fertilizer, pesticides, etc.) for every square meter in a crop field as needed at every plant growth stage[11,13,14].

There are a number of studies on agricultural digitalization using the above advanced technologies[6,8,9,11,15]. Jayaraman et al. presented an IoT-based platform, SmartFarmNet, that can automatically collect data on environments, soil conditions, fertilization, and irrigation[10]. Furthermore, it can integrate data from other sources, and all data can be stored on the cloud server to analyze and present the results to the API user. Teschner et al. showed that an UAV-based intrusion detection DT is effective in protecting agricultural fields[16]. In this study, all data was distributed over 5 G communication networks. Moghadam et al. initiated a DT at an orchard scale[17]. Specifically, they created a system that scans the status of every tree using 3D LiDAR cameras. This orchard DT enables continuous monitoring of tree's health, structure, and fruit quality, and it predicts tree's stress level, presence of disease, and yield loss. Given all information, the DT simulates various scenarios based on environmental and management parameters[17]. Delgado et al. proposed a WebGIS framework that collects geospatial data and aggregates into regional and global views of agriculture to support big data analytics for farmers and agricultural policymakers[11]. Verdouw & Kruize reported six cases of using the FIWARE open source platform for the development of agricultural DT for the first time[18]. The FIWARE easily connects IoT sensors and provides cloud services and Open APIs to enable real-time data processing and big data analysis. As an example of crop management using a deep learning algorithm, Anagnostis et al. proposed an approach for orchard tree segmentation using aerial images based on the U-net algorithm, a convolutional neural network variant[19]. This model was proven to be effective in the detection and localization of tree canopies, achieving performance levels up to 99%. Jiang et al. predicted forest change trends in the study area using a DT approach based on machine learning[20]. This DT was based on remote sensing imagery from the Landsat 7 satellite to investigate forestry canopy, species, and distribution succession. In post-harvest management, Tagliavini et al. proposed a DT that can manage the quality of harvested mangoes[21]. In this case, throughout the cold chain, computational fluid dynamics was used to evaluate quality losses, such as fruit firmness, total soluble solids, acidity, and vitamin content. As such, applications of DT have become sophisticated and diversified.

The Republic of Korea established a smart farm research policy to advance smart farm research from indoor greenhouses to open fields. As part of this, our research team aimed to develop a DT for managing mandarin (*Citrus unshiu*) orchards in open fields. Fruit crops are propagated through asexual reproduction (grafting), resulting in every tree being of the identical genomes. Additionally, as fruit crops are perennial, the data can be updated annually from the same individuals, enabling spatiotemporal analysis. Unlike other crops, fruit crops require ample space per individual, which facilitates the collection of individual-specific data.

Unfortunately, unforeseeable research policy changes with budget cuts stopped all open-field smart farm research projects. As a result, we could not continue collaborations with data producers involved in mandarin cultivation. As a surrogate, we aggregated and centralized the data that was available independently from different sources (Fig. 1). According to the Act on Promotion of the Provision and Use of Public Data[22], the Korean government has released a large number of datasets generated from various national and public institutions, and this public information is referred to as open data. Each public institution can directly provide open data generated and acquired by itself, as well as through the open data portal (https://www.data.go.kr), an integrated archive that can store and provide all open data in one site. The Rural Development Administration (RDA) annually surveys soil chemical properties according to land-use type and provides the information to farmers for field-specific fertilization. Using this information, farmers can supply an appropriate amount of fertilizer to crops at a low cost and contribute to environmental conservation by eliminating fertilizer misuse and preventing soil leakage. Jeju Island, located at the lowest latitude in the Korean Peninsula, produces mandarin fruits with a dormant volcano in the center of the island. The Jeju Free International City Development Center (JDC) collected data on fruit sugar content and size, weather information, and agricultural practices in mandarin orchards. The JDC surveyed thirty randomly selected orchards in 2021 and has made the information available through the data portal. Geocoding of the data was performed through the Kakao Developers server based on regional codes and address information published by the Ministry of the Interior and Safety (MOIS). In order to visualize the geocoded data on a map, the GIS map files were downloaded from the National Spatial Data Infrastructure Portal (NSDIP). All open data is distributed through the Open API, and it is freely and easily accessible[23].

The aim of this study is to showcase the feasibility of an agricultural DT to support data monitoring and data-driven decisions. We selected mandarin as a model crop for this study due to its wide cultivation in Jeju Island and its perennial nature. We judged that such sustainable conditions are necessary for the long-term success of DT in the future. This article illustrates that through the integration of multiple datasets obtained from diverse sources (utilizing Open APIs) and the creation of a DT for mandarin orchard management, we can achieve not only precision agriculture but also individualized agriculture, where each fruit tree is managed on an individual basis. The available datasets encompass various information including soil chemical properties, fruit quality, weather data, and agricultural practices; they are analyzed at regional, inter-orchard, and intra-orchard scales; and an interactive applet, R Shiny, is created to demonstrate how an agricultural DT can support data-driven decisions for policymakers, researchers, distributors, and farmers. It is very important to monitor at regional, inter-orchard, and intra-orchard levels, and in particular, it is essential to monitor fruit quality from individual trees on a regular basis for successful individualized agriculture, and the DT can add value by making the monitoring accurate and efficient.

## Results
### Regional scale analyses
The spatial information and observed soil components are presented in Fig. 2 using the kernel density estimation (KDE). The soil components include available phosphate, exchangeable cations (Exch. K, Ca, and Mg), acidity (pH), organic matter, and electrical conductivity. The color gradation from blue (low values) to yellow (high values) is used in the figure. This visualization accounts for locations and soil conditions observed in Jeju Island, excluding the other available information such as rice paddy and greenhouse soils from the KDE analysis. The levels of available phosphate, Exch. K and Mg, pH, and electrical conductivity tended to be higher in the western region of the island, when compared to the eastern region, in contrast to Exch. Ca and organic matter. According to the JARES report[24], the western part of Jeju Island is a non-volcanic ash soil area which is similar to the land soil of the Korean Peninsula and is highly productive. On the other hand, the eastern part is a volcanic ash soil which is characterized by low organic matter and available phosphate content and high Exch. Ca content. These soil characteristics are likely responsible for the high available phosphate content of the orchards in the western region, a non-volcanic ash soil region (Fig. 2A). Relative to the western region, the observed levels of available phosphate in the eastern region were close to 200–300 mg kg$^{-1}$ which is a range recommended by RDA. It appears that farmers in the eastern region applied a lot of organic fertilizer to address the low organic matter content due to the regional characteristics of a volcanic ash soil. As a result, the observed levels of organic matter in the eastern region were above the recommended level of 110–150 g kg$^{-1}$ (Fig. 2F). If these patterns are continuously observed and confirmed, policymakers or local government

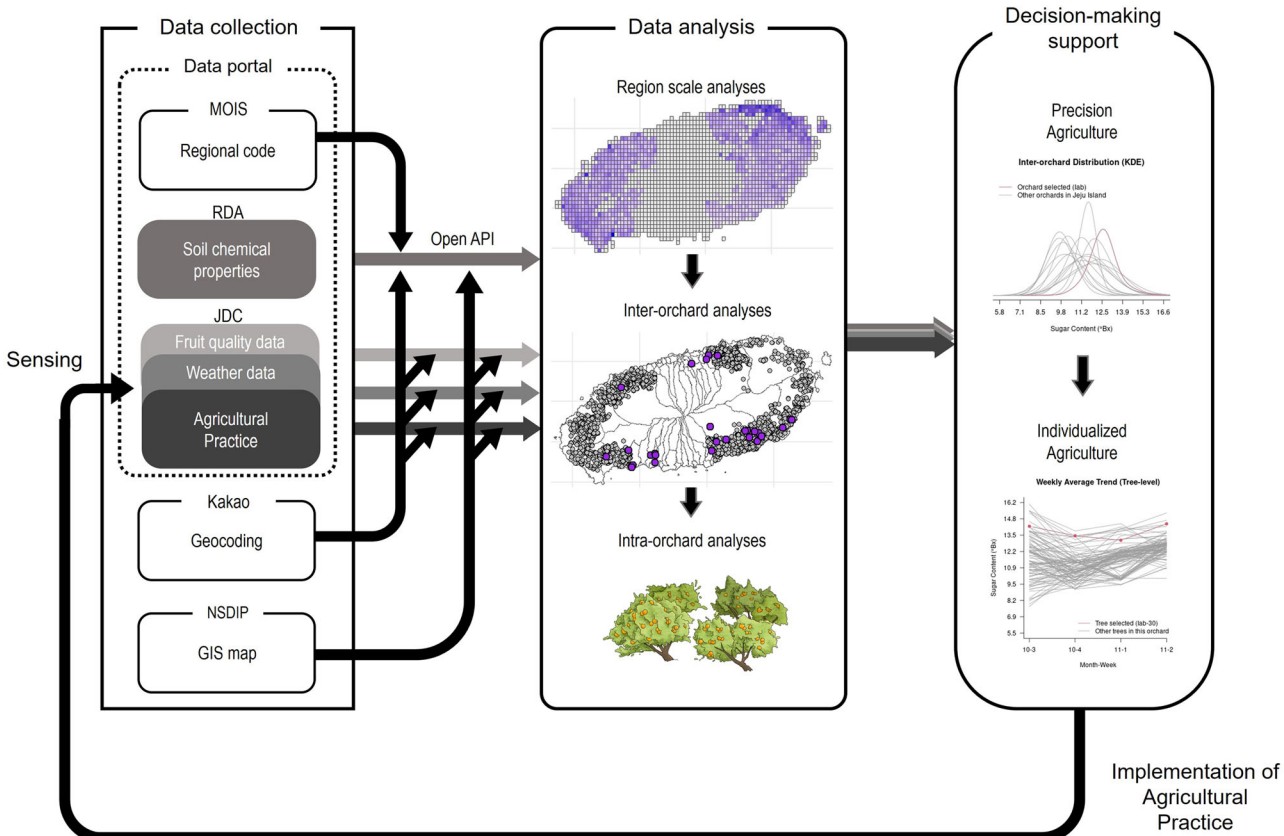

**Fig. 1 | The schematic diagram illustrates the process of data collection, analysis, and decision-making support via an agricultural digital twin.** The independent data collected from various sources through the Open API are merged and geocoded with spatial information. The merged data can be displayed on a GIS map and analyzed using various statistical methods and machine learning algorithms. The analyzed data enable regional scale, inter-orchard, or intra-orchard analysis, providing customized information based on the perspective of stakeholders. Through this process, stakeholders can receive support for regional or orchard-level decisions, and they can obtain customized information at the tree-level within an orchard. Therefore, future agricultural systems have potential to evolve from precision agriculture to individualized agriculture. On the map of Jeju Island, the gray dots indicate the location of all identified orchards, and the purple dots indicate the selected orchards presented in this paper. The abbreviations are as follows: MOIS, Ministry of the Interior and Safety; RDA, Rural Development Administration; JDC, Jeju Free International City Development Center; NSDIP, National Spatial Data Infrastructure Portal. Source data are provided as a Source Data file.

agricultural officials may plan for the supply and demand of organic fertilizer in Jeju and prepare alternatives to resolve regional imbalances. Additionally, the western region is more likely to be alkaline due to the higher soil pH, and policymakers need to encourage the supply of pH-lowering lime-based fertilizers to orchards in the western region. Currently, frequencies of soil data collection are irregular and soil data are relatively scarce (about once a year). More frequent data collection is needed to increase knowledge regarding the regional soil conditions and regional relationships with mandarin fruit quality.

The anticipated fruit size and sugar content may depend on regions and harvest time. The time- and location-specific estimates are depicted in Fig. 3. The size of each data point is proportional to the estimated average fruit size, and sugar content levels are color-coded. A darker color (brown) indicates a high sugar level (>11.5° Brix). In late October, it was uncommon to observe an average sugar level above 11.5° Brix, but it was more common in mid- and late November. During late November, the observed sugar level was higher in the orchards located in the southern region (latitude below 33.4°N) than the orchards in the northern region. It appears that the time and location served as informative layers that influence the sugar content and fruit size, respectively.

The merged dataset showed that sugar content had monotonic relationships with some soil- (Exch. K, Mg, pH, and electrical conductivity) and weather-related variables (temperature and humidity) and non-monotonic relationships with available phosphate, Exch. Ca, organic matter, and air pressure. Relatively high sugar content, rather than big fruit size, was observed near the ranges of available phosphate and organic matter recommended by the RDA (Fig. 4A, F). In contrast, Exch. Ca had a stronger relationship with fruit size rather than with sugar content (Fig. 4C). This observation suggests suppressing excessive application of Ca fertilizer in order to avoid oversized fruit. The sugar content tended to increase when air pressure is between 0 and 5 atm, while the fruit size tended to decrease. The opposite trends were observed when air pressure is above 6 atm (Fig. 4J). The inverse relationship between sugar content and fruit size was also found in the recommended ranges of available phosphate, Exch. Ca, and organic matter. As explained above, however, if variations in sugar content and fruit size were primarily influenced by the soil and weather factors, it would not be possible for farmers to artificially control these factors by altering air pressure.

## Inter-field analyses

The frequency and time of agricultural practices varied among orchards. The majority of orchards (27 out of 30) recorded and provided information on agricultural practices including pruning, fertilization, spraying, mulching, and thinning. Pruning is the process of selectively removing parts of a plant, such as branches or stems, with the aim of enhancing plant growth, increasing fruit yield, and improving overall fruit quality[25]. Pruning plays crucial roles in achieving high-quality fruit production and maintaining a consistent fruit size[26]. Additionally, it stimulates sugar content in the fruit. Optimal fertilization increases fruit production by influencing fruit weight and quality[27]. Spraying can

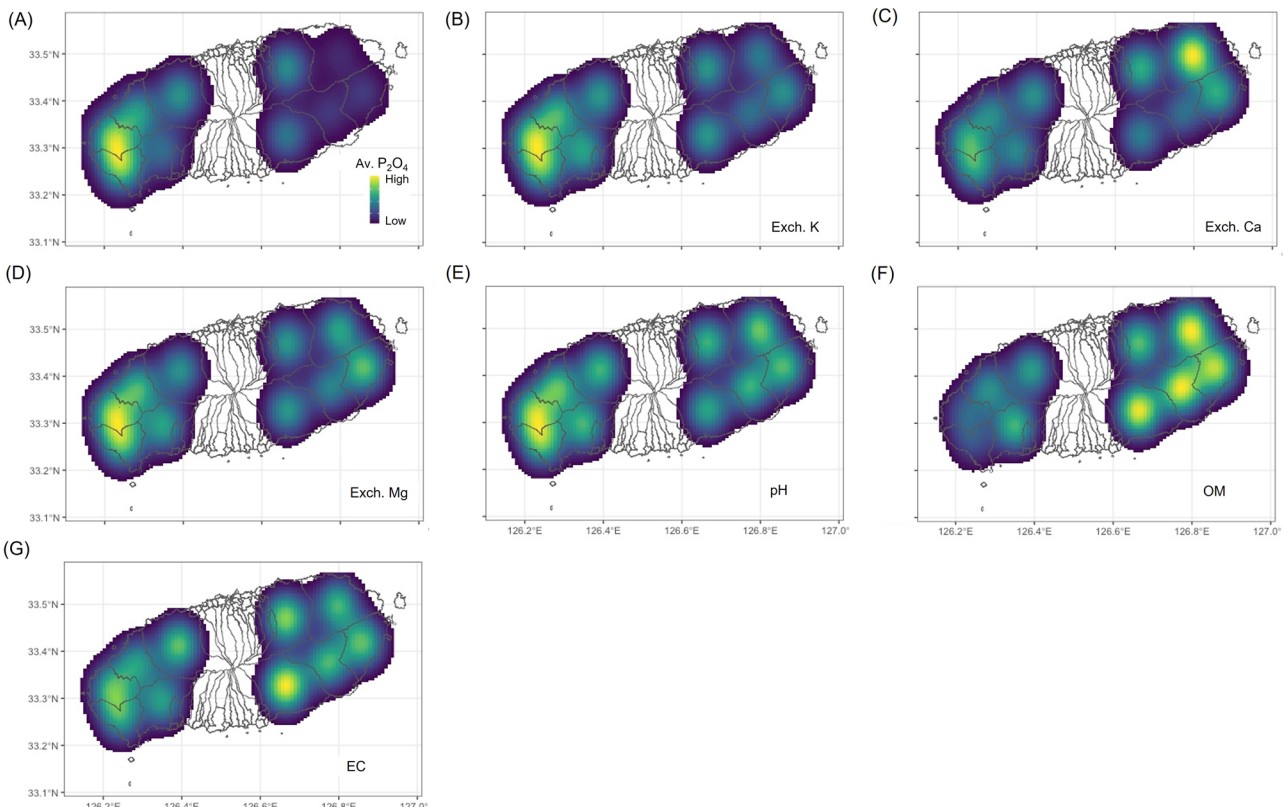

**Fig. 2 | Kernel density estimation maps of soil chemical properties with regard to mandarin orchards on Jeju Island. A**: available phosphate (Av. $P_2O_4$), **B**: exchangeable potassium (Exch. K), **C**: exchangeable calcium (Exch. Ca), **D**: exchangeable magnesium (Exch. Mg), **E**: soil acidity (pH), **F**: organic matter (OM), and **G**: electrical conductivity (EC). The estimated level of each soil chemical component near the selected mandarin orchards is represented by the color gradation from blue (low values) to yellow (high values) in each panel. Source data are provided as a Source Data file.

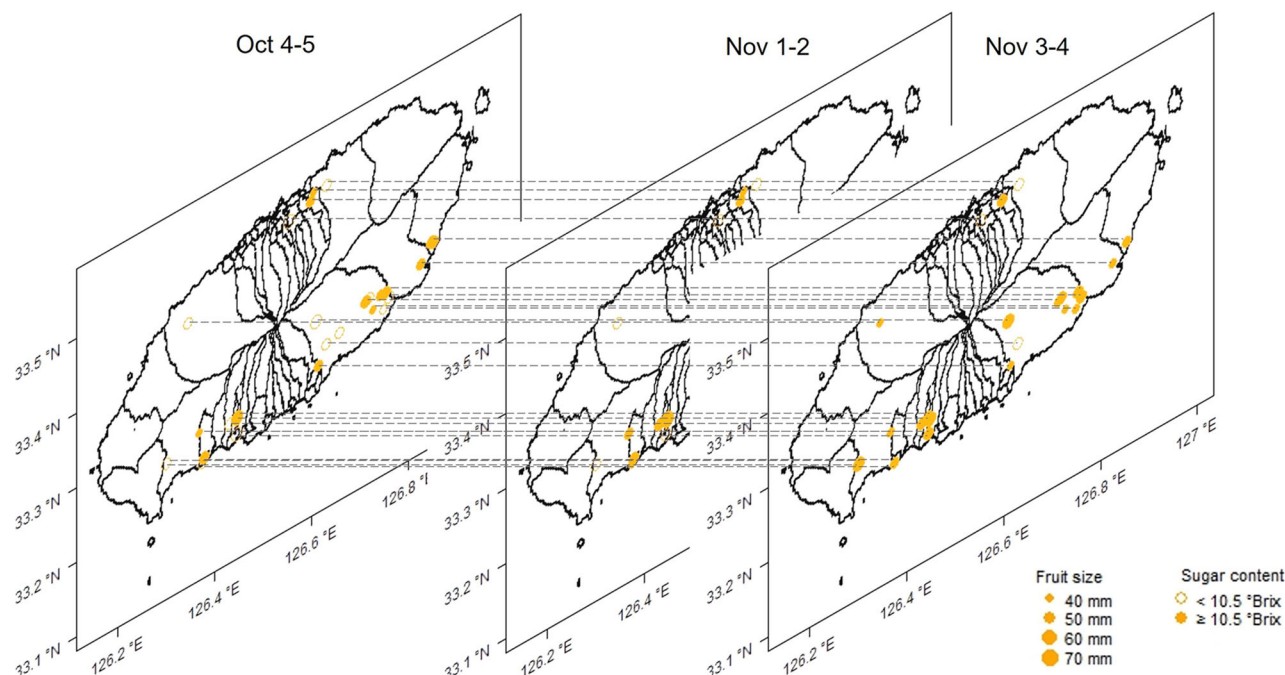

**Fig. 3 | Spatiotemporal variability of sugar content and fruit size with respect to mandarin orchards in Jeju Island.** The changes in sugar content and fruit size by orchard and time (from the fourth-fifth weeks of October to the third-fourth weeks of November) are indicated by the circle type (open and closed) and the size. With a digital twin, fruit quality data can be explained by specific location and time. The figure represents 27 selected mandarin orchards of the 30 mandarin orchards surveyed by Jeju Free International City Development Center. Source data are provided as a Source Data file.

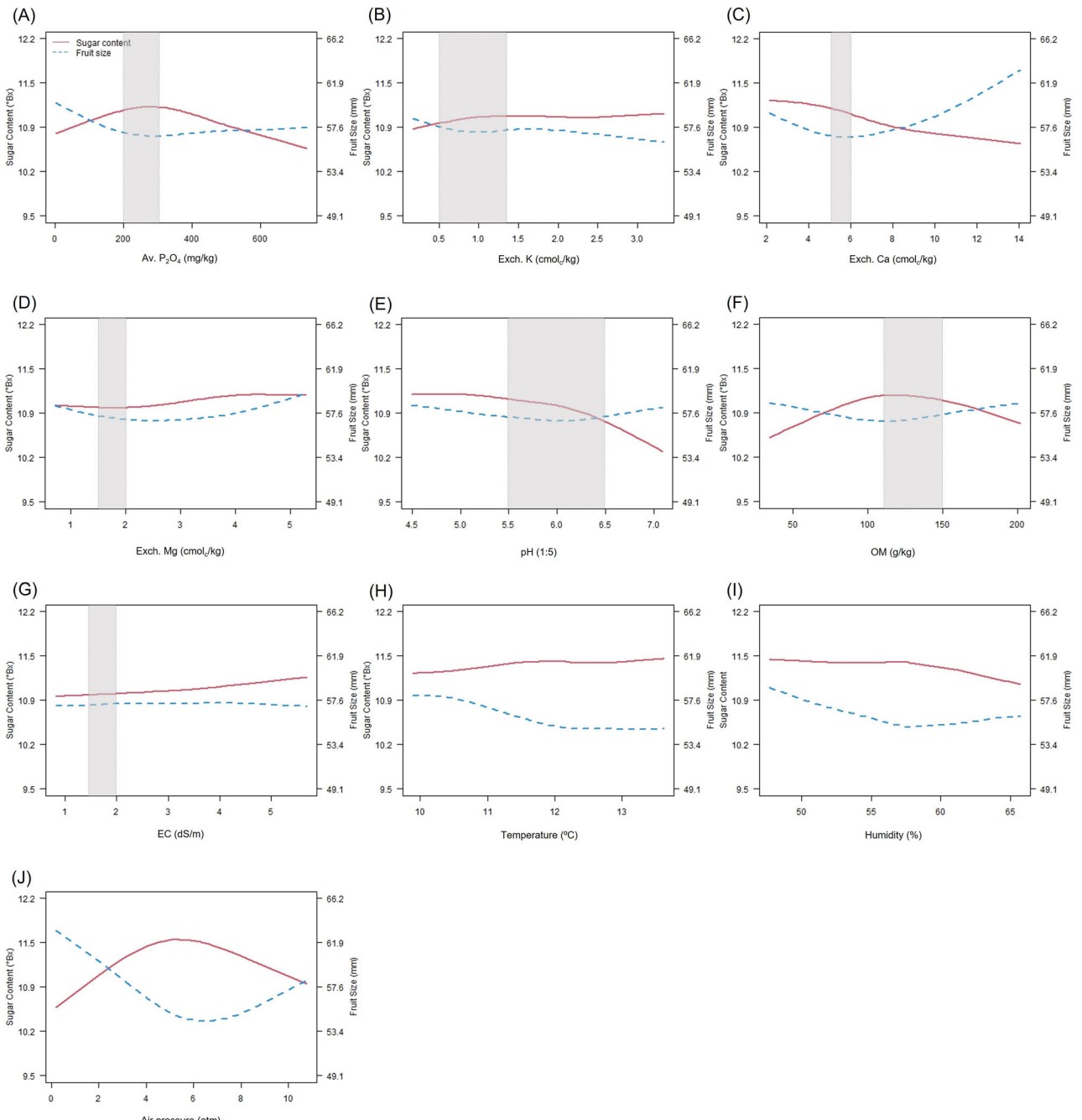

**Fig. 4 | The average trends of sugar content and fruit size with respect to soil chemical properties and weather-related conditions. A**: available phosphate (Av. P₂O₄), **B**: exchangeable potassium (Exch. K), **C**: exchangeable calcium (Exch. Ca), **D**: exchangeable magnesium (Exch. Mg), **E**: soil acidity (pH), **F**: organic matter (OM), **G**: electrical conductivity (EC)**H**: temperature, **I**: humidity, and **J**: air pressure. The gray box represents the appropriate range for each component of mandarin orchard soil recommended by the Rural Development Administration. Source data are provided as a Source Data file.

help control pests and diseases that can harm trees and lower fruit quality. Mulching serves for retaining soil moisture and suppressing weed competition which promote plant growth. Thinning is an vital agricultural practice in fruit production that is used to reduce the number of fruits per tree and improve fruit size[28]. While each agricultural practice is known to be beneficial, the observed frequency and time of application varied between orchards. Supplementary Fig. 1 presents a ridge plot displaying the monthly frequencies of agricultural practices performed by all orchards combined. The spraying started in March and was frequently done between April and September, and this kind of information allows researchers to investigate the targeted prevalent pathogens or pests. Thinning of fruits occurred

between April and September with most farmers doing it in July or August. Farmers generally made similar decisions regarding the time of mulching (mostly in June or July) and pruning (typically in March). However, the timing of fertilization showed high variability, ranging from February to July, and some farmers fertilized even in January and October. If this kind of information on agricultural practices is available over time and is associated with fruit quality, it will facilitate planning and operational decisions. Such a system will be especially beneficial for inexperienced farmers or those new to the region.

For the marketing purpose, fruit quality is categorized based on the two main factors, sugar content and fruit size. Mandarin fruits typically have a sugar content exceeding 10° Brix to be sold. Given this

condition, quality classes were established based on the fruit size as follows: too small (<49 mm), 2 S (49–54), S (54–59), M (59–63), L (63–67), 2 L (67–71), and too large (>71). Fruits that were either too small or too large were deemed unsaleable, while the other classes were suitable for the market. Fruits classified as S or M were rated as prime grade, indicating the highest quality, and fruits categorized as 2 S, L, or 2 L were rated as saleable at a fair quality. The fruit quality and soil chemical properties were compared between the two orchards, Hab in the western region and Iab in the eastern region, which growed the same cultivar named Miyagawa Wase. The comparison revealed significant differences in soil chemical properties which could be associated with the variations in fruit quality (Fig. 5 and Table 1). The gray box in Fig. 5A represents the recommended fertilization standards by RDA[29]. The Iab orchard had a low level of available phosphate content and a high level of Exch. Ca content due to the characteristics of volcanic ash soil found in the eastern region. Most orchards had adequate levels of organic matter due to government policies aiming at increasing the supply of organic fertilizer. However, unlike the Iab orchard, the Hab orchard had a high level of electrical conductivity which indicates an excess of nutrients in the soil that can negatively impact the nutrients uptake by the mandarin tree. The Iab orchard performed the agricultural practices in later months than the Hab orchard did with an exception of pruning (Fig. 5B). The sugar content of Iab was higher, except in late October, and the fruit size of Iab was larger, except in early November (Fig. 5C), when compared to those of Hab. The fruit sugar content from both orchards exceeded the marketable standard of 10° Brix. However, if this pattern persists annually, it is advisable to harvest from the third week of October to the first week of November for the Iab orchard and from the first week of November to the fourth week of November for the Hab orchard (Fig. 5D) in order to obtain prime grade fruit (54–63 mm).

Significant variations in fruit ripening and quality were observed between the two orchards, even though they were growing the same cultivar, indicating the identical genotype. These differences can be attributed to distinct environmental factors and variations in agricultural practices and management as discussed earlier. Similarly, notable variations in sugar content and fruit size were observed within the same orchard (Fig. 5C), despite growing the same genotype. These variations on a smaller scale are likely influenced by micro-environmental factors. This clearly emphasizes the importance of collecting micro-environmental data within the orchard. Such data will provide information on the environment × management interaction for researchers and profits for farmers and distributors. They can sell and purchase high-quality fruits from each orchard at optimal times to maximize their profits.

There were different trends in sugar content and fruit size among orchards. The inter-orchard variation in fruit size, where the median size ranged from 43.9 to 67.6 mm, was significantly greater than the inter-orchard variation in sugar content, which ranged from 9.8 to 12.1 °Brix (Fig. 6). Unlike sugar content, the orchard with the lowest median fruit size (Xab) and the orchard with the largest median fruit size (Uab) were clearly distinguishable (Fig. 6B). The Xab orchard produced fruits of relatively high sugar content, whereas the Uab orchard produced fruits of relatively low sugar content (Fig. 6A). As aforementioned in the regional scale analysis (Fig. 4), it seems that sugar content and fruit size are inversely related in mandarin fruit. To this end, the determination of harvest time is very important to balance the two mandarin quality factors, which eventually increase the proportion of prime grade fruit.

Additionally, the trend of sugar content increased over time in most orchards, while the trend in fruit size did not follow a consistent pattern across orchards (Supplementary Fig. 2). Mandarin fruit was harvested simultaneously starting from the third week of October, coinciding with the beginning of fruit quality surveys. The mixed-effects model estimated that the mean sugar content continued

increasing over time, largely unaffected by ongoing harvesting, and the continual increase of the mean sugar content was strongly significant ($p < 0.001$). When compared to the initial week, the third week of October (denoted by 10-3), the estimated mean sugar content was higher by 0.309, 0.308, 0.658, 1.047, 1.248, and 1.435 °Brix for 10-4, 10-5, 11-1, 11-2, 11-3, and 11-4, respectively. The model estimated that the mean fruit size decreased after the initial week, but the pattern was not as clear as the mean sugar content. See Supplementary Table 1 for the estimated parameters under the mixed-effect model. The time factor is limited to soil, weather, and agricultural practices in our research data. When the same cultivar is planted, variations in the longitudinal trend of fruit quality between orchards can be attributed to different environments and management.

The automatic machine learning (AutoML) algorithm, implemented by the 'h2o' package in R[30], predicted the fruit size better than the sugar content when they were analyzed with the time of fruit observation and orchard-level variables including the five agricultural practices (fertilization, mulching, pruning, spraying, and thinning) and the three weather variables (temperature, humidity, air pressure). The selected model was the stacked ensemble model which resulted in root mean square error (RMSE) = 0.97, mean absolute error (MAE) = 0.76, and R-square ($R^2$) = 0.43 for sugar content and RMSE = 3.73, MAE = 2.96, and $R^2$ = 0.84 for fruit size. The orchard index was identified as the most important predictor followed by air pressure for both sugar content and fruit size. At the orchard-level, the prediction of fruit size is substantially better than of sugar content (Fig. 7), and it implies that intra-orchard analysis may be needed especially to improve the sugar content prediction.

## Intra-field analyses

Significant variability in fruit quality was observed within orchards. In addition to differences between orchards, it is crucial to comprehend the variability of fruit quality within orchards, known as intra-orchard differences. For illustrative purposes, the Iab orchard is selected in Fig. 8. In the dataset, fruit samples were categorized into high, middle, and low positions based on their height from the ground level, and three samples were collected from each position per tree each week. The distributions of sugar content and fruit size seemed to be quite similar across the three position levels (Fig. 8A).

When analyzed with the mixed-effect model, there was a difference in expected sugar content based on the position, but the estimated difference in average sugar content was minimal. Fruits in the high position had higher sugar content than those in the middle position only by 0.020 °Brix on average ($p = 0.131$), and fruits in the middle position had higher sugar content than those in the low position by 0.039 °Brix on average ($p = 0.000251$). However, there were no statistically significant differences in fruit size among the three positions. Therefore, grading fruits based on their position appears to have little practical significance. See Supplementary Table 2 for the estimated parameters under the mixed-effect model.

Hierarchical clustering analysis grouped individual trees in the Iab orchard into four clusters based on the observed trends of sugar content and fruit size over time (Fig. 8). One of the clusters (Cluster 2 in Fig. 8A) clearly shows an increasing trend of sugar content during the survey period, whereas the other three clusters do not. On the other hand, all of the four clusters show decreasing trends of fruit size (Fig. 8B). There was substantial variation in sugar content among trees over time, indicating potential room for improvement in sugar content with specific management practices, even when the same cultivar is planted in the same orchard. For instance, after identifying a group of trees producing fruits with low sugar content through hierarchical clustering analysis (Cluster 3 in Fig. 8A), customized agricultural practices can be applied to each selected tree to improve its sugar content. The groups were categorized based on sugar content in the third week of October, when the investigation began. Trees with high

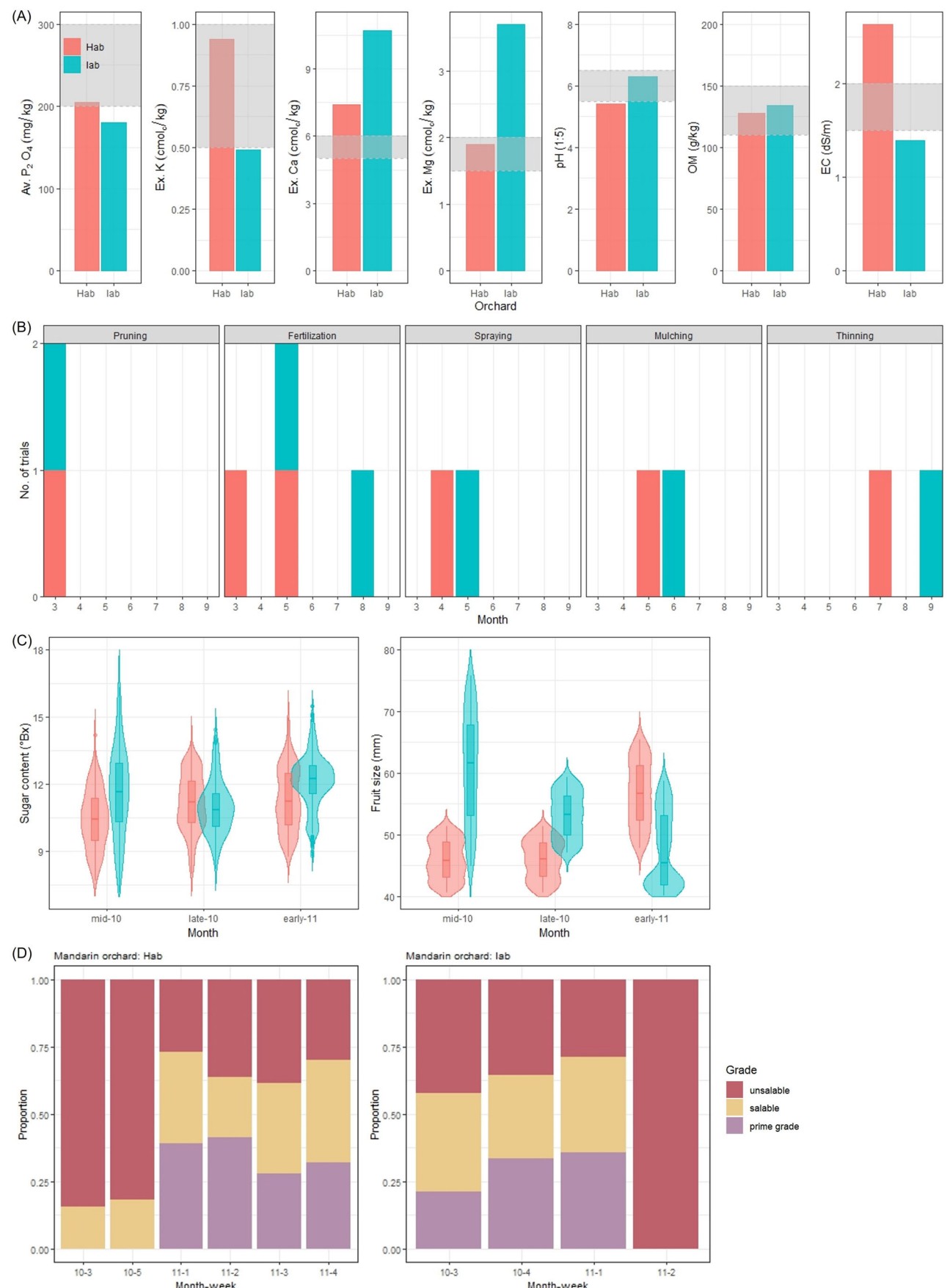

**Fig. 5 | Inter-orchard analysis between the two orchards, Hab (located in the eastern region) and Iab (western region).** The soil chemical properties of the two orchards are compared with the fertilization standard (gray box) recommended by the Rural Development Administration (**A**). The monthly frequency of each agricultural practice is compared between the two orchards, and Comparison of monthly number of agricultural practices performed in each orchard (**B**). The violin plots compare the distributions of sugar content and fruit size in the two orchards over time; $N = 300$ mandarins in mid-10, late-10, and early-11 for Hab; $N = 300$ in mid-10 and late-10 300 and $N = 600$ in early-11 for Iab. The 5-number summaries used for the boxplots (minimum, first quartile, median, third quartile, and maximum) of sugar content (°Bx) are (7.7, 9.5, 10.5, 11.4, 14.2) in mid-10, (6.8, 10.3, 11.2,

12.1, 13.9) in late-10, and (8.8, 10.2, 11.3, 12.5, 15.0) in early-11 for Hab; and (7.2, 10.3, 11.7, 12.9, 16.3) in mid-10, (8.2, 10.1, 10.9, 11.6, 14.4) in late-10, and (8.8, 11.6, 12.2, 12.8, 15.5) in early-11 for Iab. The 5-number summaries of fruit size (mm) are (40.6, 43.2, 45.8, 48.8, 51.4) in mid-10, (40.6, 43.3, 46.1, 48.7, 51.4) in late-10, and (47.9, 52.4, 56.7, 61.3, 65.5) in early-11 for Hab; and (44.9, 53.2, 61.6, 67.8, 75.8) in mid-10, (47.1, 49.9, 53.2, 56.3, 59.4) in late-10, and (40.1, 41.9, 45.4, 53.1, 58.7) in early-11 for Iab (**C**). The barplots show the proportions of unsaleable, saleable, and prime grade were compared over time; $N = 300$ mandarins in 10-3, 10-5, 11-1, and 11-2 and $N = 288$ mandarins in 11-3 and 11-4 for the Hab orchard; $N = 300$ mandarins in 10-3, 10-4, 11-1, and 11-2 for the Iab orchard (**D**). Source data are provided as a Source Data file.

sugar content in the third week of October maintained high sugar content until the second week of November. In contrast, trees with initially low sugar content retained low sugar content until the second week of November. Therefore, trees with low sugar content may benefit from tailored agricultural practices such as rain-shelter cultivation, irrigation control, foliar fertilization, proper pruning, and thinning to enhance their sugar content. Similarly, when clustering analysis identifies trees with very small fruit sizes, simultaneous agricultural practices like fruit thinning or late harvesting can be applied. As such, clustering analysis can be another statistical tool for

## Table 1 | The comparison between the Hab and Iab orchards in Jeju Island

| | | Hab (western region) | Iab (eastern region) |
|---|---|---|---|
| Location | Latitude (°N) | 33.28 | 32.33 |
| | Longitude (°E) | 126.37 | 126.74 |
| Soil Component | Available phosphate (mg/kg) | 205 | 180 |
| | Exchangeable potassium (cmol$_c$/kg) | 0.94 | 0.49 |
| | Exchangeable calcium (cmol$_c$/kg) | 7.4 | 10.7 |
| | Exchangeable magnesium (cmol$_c$/kg) | 1.9 | 3.7 |
| | Soil acidity (1:5) | 5.4 | 6.3 |
| | Organic matter (g/kg) | 128 | 134 |
| | Electrical conductivity (dS/m) | 2.64 | 1.39 |
| Agricultural Practice | Pruning (month) | Mar | Mar |
| | Fertilization (month) | Mar, May | May, Aug |
| | Spraying (month) | Apr | May |
| | Mulching (month) | May | Jun |
| | Thinning (month) | Jul | Sep |
| Fruit Quality | Mean (SD) of sugar content (°Brix) | Mid-Oct: 10.4 (1.31) Late Oct: 11.2 (1.29) Early Nov: 11.4 (1.39) | Mid-Oct: 11.7 (1.97) Late Oct: 11.0 (1.19) Early Nov: 12.1 (1.17) |
| | Mean (SD) of fruit size (mm) | Mid-Oct: 46.0 (3.15) Late Oct: 46.0 (3.10) Early Nov: 56.7 (5.17) | Mid-Oct: 60.9 (8.82) Late Oct: 53.1 (3.65) Early Nov: 47.5 (6.04) |
| | Prime grade (%) | Mid-Oct: 0.0 Late Oct: 0.0 Early Nov: 39.0 | Mid-Oct: 21.3 Late Oct: 33.7 Early Nov: 35.7 |
| | Unsalable (%) | Mid-Oct: 84.3 Late Oct: 81.7 Early Nov: 27.0 | Mid-Oct: 42.3 Late Oct: 35.7 Early Nov: 28.7 |

individualized agriculture by determining specific needs of individual trees to improve sugar content and fruit size.

The orchard-level mixed-effects model demonstrates the potential of individualized agriculture within each orchard. When considering the harvest time and the five orchard-level agricultural practices only, 19% of the variance in sugar content was explained ($R^2 = 0.19$). However, the orchard-level mixed-effects model explained 38% of the variance in sugar content ($R^2 = 0.38$), indicating that the current orchard-level practices have limited capacity to explain the variation in sugar content between orchards. On the other hand, when the tree-level model was applied, 66% of the variance in sugar content was explained ($R^2 = 0.66$). Figure 9 displays the scatterplot of predicted and observed values of sugar content, highlighting the different predictive powers between inter-field analysis (Fig. 9A) and intra-field analysis (Fig. 9B). It appears that the predictive power of the intra-field analysis is better than of the inter-field analysis which implies that farmers can benefit from tree-level management (individualized agriculture) in addition to the orchard-level management (precision agriculture), and an agricultural DT can be a helpful tool for monitoring individual trees in an orchard. As such, the development of DT has a potential to open up the transition from precision agriculture to individualized agriculture. All supplementary data used to create all figures (Fig. 1 to 9) and supplementary figures (Supplementary Fig. 1 and 2) provided in Supplementary Data.

### Agricultural DT demonstration

For the purpose of demonstration, a tree of the Iab orchard is demonstrated using the agricultural DT (https://stevenkimcsumb.shinyapps.io/ShinyDT/). In the webpage, the user may select the orchard (Iab), click on the Submit button below, select the fifth tree (Iab-5), and click on the Submit button below. After the submission, there are nine panels shown: Map, Soil, Weather, Agricultural Practice, Sugar Content Distribution, Fruit Size Distribution, Sugar Content History, and Fruit Size History. The Map panel presents the location of the Iab orchard in Jeju Island. The Soil panel presents the percentile of each soil component (available phosphate, exchangeable potassium, exchangeable calcium, exchangeable magnesium, soil acidity, organic matter, electric conductivity) compared to other areas in the island. It also shows whether the observed level is within the RDA recommendation or not. The Weather panel presents the percentile of temperature, humidity, and air pressure compared to other areas in the island. The Agricultural Practice panel compares the five agricultural practices (fertilization, mulching, spraying, pruning, and thinning) to other orchards in the island. The Sugar Content Distribution and Fruit Size Distribution compare the sugar content and fruit size, respectively. In each panel, the Iab orchard is compared to other orchards in the island, and the Iab-5 tree is compared to other trees in the Iab orchard. The Sugar Content History and Fruit Size History show the weekly patterns of sugar content and fruit size, respectively, at orchard-level and tree-level. The DT will show that the average sugar content and fruit size increases and decreases, respectively, with respect to time at both orchard-level (Iab) and tree-level (Iab-5).

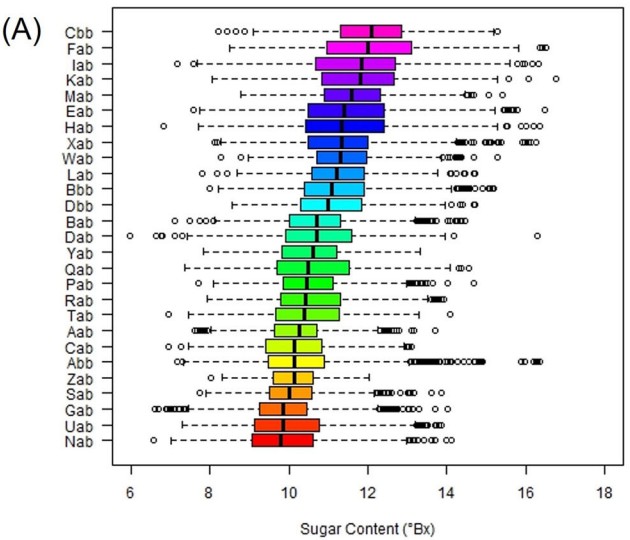
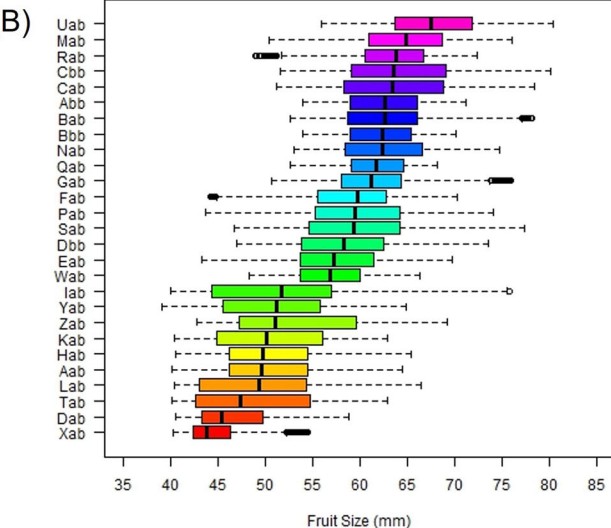

**Fig. 6 | The distributions of sugar content and fruit size by orchard. A**: sugar content, **B**: fruit size. The orchards are ordered by the median sugar content (**A**) and the median fruit size (**B**) among $N = 39{,}679$ fruits observed between mid-October (the third week of October) and late November (the fourth week of November) in the 27 orchards. The Nab orchard has the lowest median sugar content (°Bx), and the 5-number summary of its boxplot (minimum, first quartile, median, third quartile, and maximum) is (6.6, 9.1, 9.8, 10.6, 14.1). The Cbb orchard has the highest median sugar content, and the 5-number summary of its boxplot is (8.2, 11.3, 12.1, 12.9, 15.3). The Xab orchard has the smallest median size (mm), and the 5-number summary of its boxplot is (40.3, 42.3, 43.9, 46.3, 54.5). The Uab orchard has the largest median size (mm), and the 5-number summary of its boxplot is (55.9, 63.7, 67.6, 71.8, 80.4) where the minimum size observed Uab (55.9 mm) is greater than the maximum size observed Xab (40.3 mm). Source data are provided as a Source Data file.

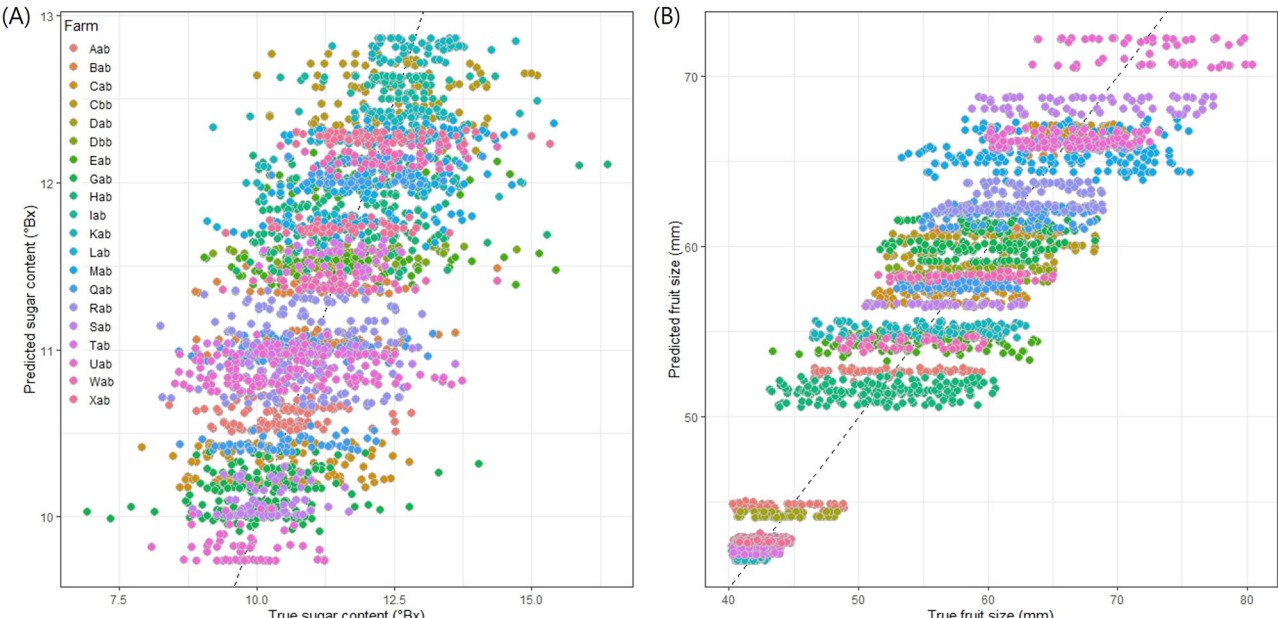

**Fig. 7 | The scatterplot of the predicted and observed sugar content and fruit size using machine learning models based on weather and fruit quality data. A**: sugar content, **B**: fruit size. Source data are provided as a Source Data file.

## Discussion

### Digital twin for soil management

Incorporating detailed soil information into regional monitoring for precision agriculture would be highly beneficial. One promising approach is the utilization of satellite image analysis proposed by ref. 31. This method involves the analysis of data obtained through remote sensing, allowing for the visualization of the distribution of individual soil components on a field-by-field basis. It can also provide critical soil information at a fine scale. They demonstrated that a near-infrared spectroscopy technology can efficiently measure nitrogen, phosphorus, and potassium in soil. This non-destructive method can resolve disadvantages associated with conventional destructive methods for soil measurement. Furthermore, it can be combined with image data from UAV or satellites to streamline and expedite data acquisition process. For instance, field-scale soil moisture maps have been generated by utilizing data from Landsat[8,32], including the normalized difference vegetation index and land surface temperature. Additionally, yield prediction models have been established by integrating soil moisture and maize yield data[33]. Using orchard-level soil chemical information from satellite imagery, it will be feasible to

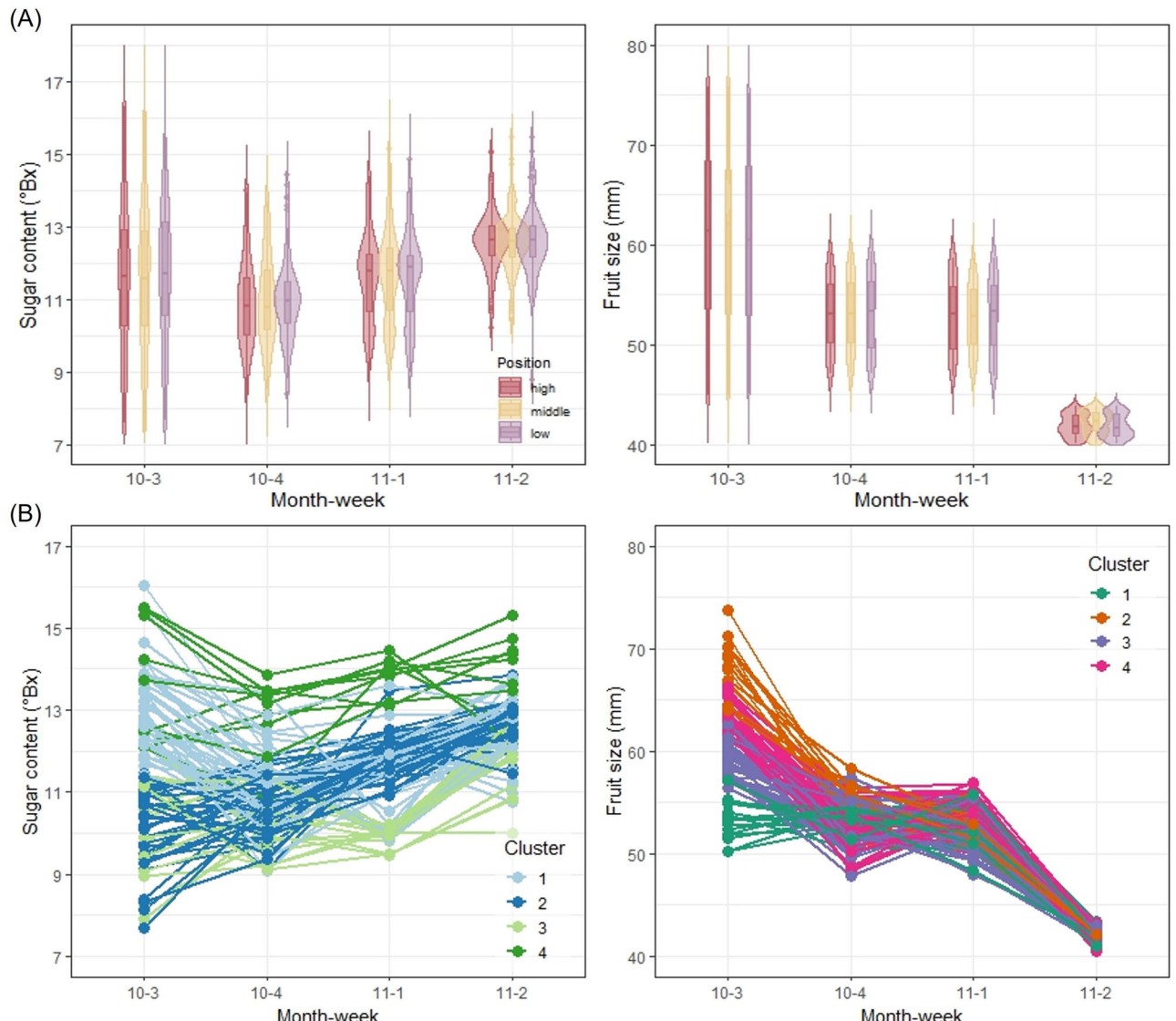

**Fig. 8 | Intra-orchard analyses of the lab orchard.** The distribution of sugar content and of fruit size are compared by the position of the fruit on the tree with respect to time; 3 mandarins per tree (one in each position) and 100 trees. The distributions are similar across the three positions, but they are different with respect to time. Combining all positions, the 5-number summaries for the boxplots (minimum, first quartile, median, third quartile, and maximum) of sugar content (°Bx) are (7.2, 10.3, 11.7, 12.9, 16.3) in 10-3, (82, 10.1, 10.9, 11.6, 14.4) in 10-4, (8.9, 10.7, 11.8, 12.3, 15.1) in 11-1, and (8.8, 12.2, 12.6, 13.0, 15.5) in 11-2; and the 5-number summaries of fruit size (mm) are (44.9, 53.2, 61.6, 67.8, 75.8) in 10-3, (47.1, 49.9, 53.2, 56.3, 59.4) in 10-4, (46.9, 50.0, 53.1, 55.8, 58.7) in 11-1, and (40.1, 41.1, 41.9, 43.1, 43.9) in 11-2 (**A**). The hierarchical clustering analysis of sugar content and fruit size (longitudinal observations of the 100 trees) is shown using 4 clusters (**B**). Source data are provided as a Source Data file.

develop a DT which can generate regional and orchard-level maps of soil profile and predict fruit quality.

Fruit trees occupy a larger space than other crops, and soil components vary within the space of an orchard. A fruit tree is directly affected by the soil underneath the tree rather than all soil in the entire field. Furthermore, achieving uniform fertilizer distribution within the two-dimensional space of an orchard is challenging, and it becomes even more complex to discern variations in fruit quality based on the regional or orchard-level soil chemical components. If we could measure the soil chemical components at specific points using an IoT sensor, it would be feasible to apply fertilization tailored to each component's needs. This means that each tree would receive the precise amount of soil nutrients necessary for its optimal growth, and point-specific fertilization would facilitate precision agriculture and precision conservation by minimizing fertilizer misuse.

A map representing the soil chemical properties should provide interactive soil and fruit information on the website for each point in the field, along with spatial information for stakeholders (e.g., policy-makers, researchers, farmers). The GIS serves as a fundamental framework for analysis of all data in relation to location. Looking ahead, there is an opportunity to integrate spatial information with data on fruit quality, weather conditions, and agricultural practices, including activities like spraying, pruning, and thinning. This integration provides regional-level, orchard-level, and tree-level information.

### Digital twin for agricultural practice management

Agricultural practices have a direct and significant impact on fruit quality. Unlike certain environmental factors, these practices are under human control and can be implemented throughout the year. As more data accumulates, the monthly records of agricultural practices become increasingly valuable. Moreover, information on agricultural practices from neighboring orchards of a specific orchard is more relevant than information from distant areas. This localized information is rooted in the local environment and accounts for inter-field

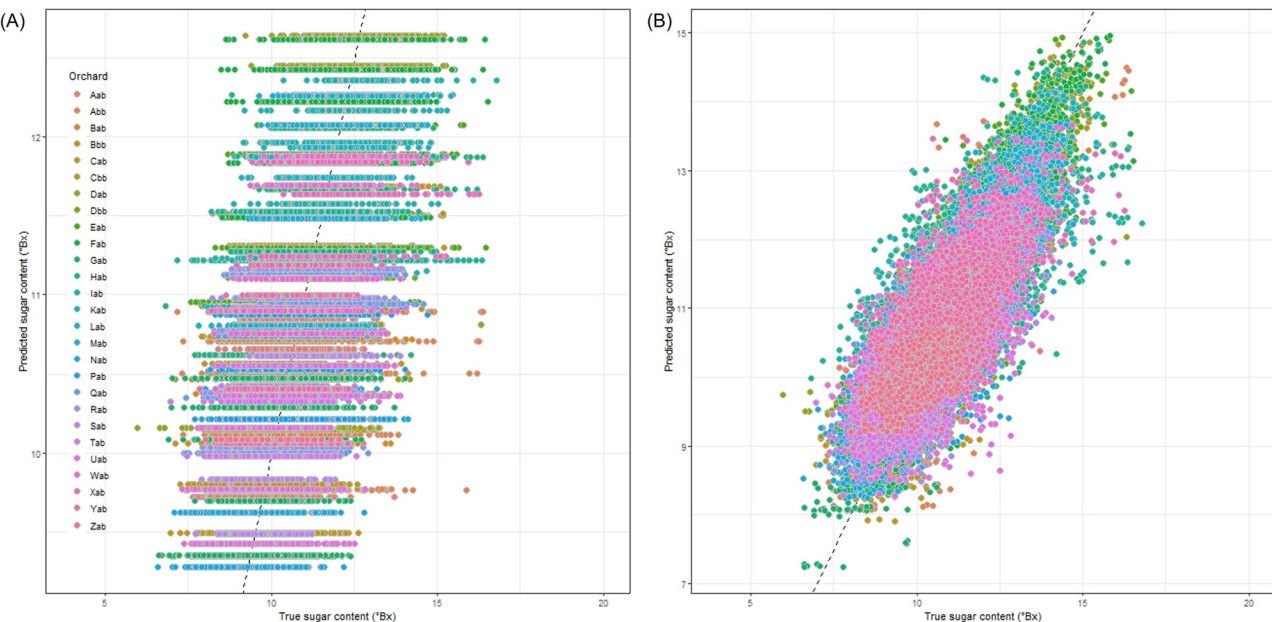

**Fig. 9 | The scatterplot of the predicted sugar content against true (observed) sugar content.** The inter-field mixed-effect model results in $R^2 = 0.380$ (**A**). The intra-field mixed-effect model results in $R^2 = 0.662$ (**B**). Source data are provided as a Source Data file.

variations. Spraying data, in combination with spatial and weather data, can serve as an effective disease and pest monitoring system. Since the spread of fungal diseases is closely linked to weather conditions, such as raindrops and wind[34,35], this system can be used to predict disease outbreaks and facilitate preventive measures. By closely monitoring and analyzing time-series data on disease outbreaks in conjunction with spatial information, it may be possible to lower the likelihood or reduce the loss due to disease spread. The important initial step is to establish automated systems for frequent data collection and management for all orchards.

The significance of agricultural practices can be assessed by increased fruit yield. For instance, pruning plays a crucial role in shaping trees and promoting flower bud differentiation, directly influencing fruit production. By meticulously organizing pruning-related data (such as pruning techniques, location of flower buds, by-product weight resulting from pruning, etc.) and metadata for survey items, the correlation between pruning and fruit production can be thoroughly analyzed. Furthermore, this approach allows for a comprehensive investigation into yield variations among different cultivars based on pruning methods. As a result, the functionality of an agricultural DT can be leveraged to create optimal environmental conditions tailored to specific cultivars.

## Digital twin for weather information management

Weather information plays a crucial role in agricultural practices and fruit quality. For example, flower or fruit thinning should commence during the flowering period, and it is important to avoid rains when thinning chemicals are applied. Decisions on the timing of agricultural practices should be made based on weather information, and it can be equipped in a DT. Moreover, it is pivotal to develop various weather metrics and metadata to study their influence on specific targets, such as fruit quality, based on the crop species. This is necessary because each crop has its unique optimal environmental conditions.

## Digital twin for fruit quality management

The overarching goal is to increase the fruit quality, but too many factors are associated with it. As presented in this study, fruit quality significantly varies between orchards and within orchards (e.g., the Iab orchard in Fig. 8). In particular, given the regular data updates, the

tree-level variations and the longitudinal patterns can be monitored for the purpose of quality control. In this sense, individualized agriculture will become a feasible agricultural system in the future. The current agricultural system, which produces high-quality products in small areas, such as greenhouses, can be expanded and applied to large areas of open fields in the future. The application of individualized agriculture is essential not only for the production of high-quality agricultural products in large areas but for multi-variety small-scale production systems as well. Recently, space agriculture has been explored in the context of space development, and the application of individualized agriculture is fundamental to produce agricultural products in a limited space.

An agricultural DT must be a shared tool for stakeholders. It requires active discussion among stakeholders, updates on a regular basis, and continuous improvements for helpful real-world feedback. For policymakers, data visualization like Fig. 2 can guide their regional decisions. Such information can be reflected in the budget of policymakers who are in charge of agricultural soils and fertilizer supply in Jeju Island. For distributors, data visualizations like Figs. 6, 7, and 8 can support inter-orchard quality assessment and tracking. If more detailed information is available than presented in this study, a DT can provide information regarding yield forecasts, expected profits, and management of distributed fruit. For researchers, data visualization like Figs. 3 and 4 can help understand regional and inter-orchard variations, generate and test hypotheses, and make practical suggestions. They can even plan matched studies by cultivar and conduct cultivar-specific studies. Finally, for farmers, Fig. 8 can help assess intra-orchard fruit quality and monitor longitudinal patterns after performing particular agricultural practices, and a DT can present detailed information at tree-level.

Improving or maintaining high fruit quality is both science and art, and farmers shall balance between empirical evidence and farmers' observations, experiences, and knowledge. At this point, it is an open question whether an individualized tree-level management will be more profitable than regional or orchard-level management. We need to consider how to lower the cost of implementing DT. The magnitude of benefits from implementing DT is unknown as of now, and we need a scientific approach to this question. We need to compare current regional or orchard-level practice versus new individualized

agriculture guided by DT using a controlled randomized experiment. Dividing orchards into the control zone and experimental zone, it is necessary to confirm and estimate the benefit of DT. This study is limited to mandarin fruit with observational data, but we want to observe and experiment with more kinds of fruit. Unlike the current speed of technological advances, it will be a patient process.

Applying individualized agriculture to cereal or vegetable crops is very challenging. Because individual plants of cereal or vegetable crops are not genetically identical, it is difficult to expect the same quality or yield under the same environment or management. In contrast, fruit crops (like mandarins) are relatively easy to study at individual levels and feasible to apply individualized agriculture because each tree propagates through asexual reproduction and have identical genomes. Thus, when researching orchards that cultivate a single cultivar (genotype), the variations in the phenotypes they exhibit are influenced by the environment and agricultural practices (management). Researching this genotype × environment × management interaction remains highly challenging, and more experimentations are needed to address this complex question. From the consumers' perspective, regardless of the scientific merit of individualized agriculture, most consumers would not purchase a ten-dollar high-quality mandarin in South Korea and elsewhere. Future studies should address lowering the cost and labor in data collection, precision agriculture, and individualized agriculture.

The long-term objective of our research team is to make regional, inter-orchard, and intra-orchard information more complete and accessible through interactive digital platforms, tailored to the goals and needs of stakeholders. We are currently developing a streamlined process that automatically retrieves data via an Open API, securely stores it on a cloud server, conducts comprehensive analyses, and disseminates the results to various IT devices. As an initial step, we have developed a freely accessible webpage (https://stevenkimcsumb.shinyapps.io/ShinyDT/) for demonstration purposes based on all information currently available to us. This version of DT does not demonstrate how to automatically suggest agricultural practices and how to assess the effect of the agricultural practices on fruit quality, and it is a main limitation of our study. The current form of interactive applet is to be improved over time and communicated with the stakeholders in order to operate in a closed loop. We anticipate that the agricultural DT opens a new era of individualized agriculture via interdisciplinary collaboration among agricultural researchers, farmers, statisticians, software engineers, and more.

## Methods

### Data resources and collection through Open API
Publicly accessible Open APIs, which are potential sources for developing an agricultural DT for mandarin orchard management, were collected from multiple sources on the data portal (https://www.data.go.kr). All processes including data collection, parsing, and analysis were performed using statistical software R[36]. The regional codes were obtained from the MOIS, soil data were obtained from the RDA of Republic Korea, and data on fruit quality, weather, and agricultural practices of mandarin orchards were obtained from the JDC (Fig. 1).

The soil data analyzed from 2020 to 2022 were collected by the administrative district of Jeju using 'xml'[37] and 'jsonlite'[38] packages in R. The chemical properties of 30,261 agricultural soils in Jeju Island were crawled, and 5939 orchard soils were used for our analysis. The soil data included available phosphate, Exch. K, Ca, and Mg, pH, organic matter, and electrical conductivity.

The JDC randomly selected the 30 mandarin orchards in Jeju and collected data on weather, agricultural practices, and fruit quality from each orchard. The weather data were obtained by installing a sensor at each orchard which is capable of recording temperature,

relative humidity, and air pressure (daily average). The data on agricultural practices were self-reported by the farmers, and they reported practice type, treatment amount, date, units, and agrochemical product name. One hundred mandarin trees were randomly selected in each orchard, and these trees were repeatedly observed from the third week of October to the fourth week of November in 2021. The sugar content (°Brix), fruit size (mm), and fruit position (high, middle, and low) were recorded with three replicates (one per fruit position level) each week per tree. These fruit quality data were measured manually by investigators using destructive measuring methods. All fruit-level information (sugar content, fruit size, and position) was matched with the month, week, day, and tag number (tree identification number). See the Data Availability section for more information.

### Data parsing
Geocoding is a computation process of converting address information into geographic coordinates (latitude and longitude), and it can be used to map locations. All address information in Jeju Island associated with the soil data was integrated into one file for the geocoding. All data were geocoded to add the spatial information through the server developed by Kakao, and all data were merged using R packages including 'rjsonio'[39], 'data.table'[40], 'dplyr'[41], and 'httr'[42].

A single data frame was created by merging the geocoding data and the soil data, and it was used for map visualization using R packages including 'terra'[43], 'maps'[44], 'sp'[45], and 'sf'[46]. For this map visualization, the GIS maps (shape files) provided by the NSDIP were used (Fig. 1). The regional-level, orchard-level, and tree-level data were merged and used for inter-orchard analysis (e.g., variation between orchards) and intra-orchard analysis (e.g., variation between trees within the same orchard). Combining all data, an agricultural DT was created with the four kinds of information: soil, weather, agricultural management practice, and fruit quality.

### Regional-scale data visualization and analysis
For the regional-scale data visualization, a 1-km grid map was created by a shape file using the QGIS program (v 3.26.2)[47], and it was combined with the soil data (available phosphate, Exch. K, Ca, and Mg, pH, organic matter, and electrical conductivity) after averaging observed values of each soil chemical component within each grid (Fig. 2A–G). The KDE was applied to describe the regional level of each soil component. The kernel-smoothed intensity function could be estimated given soil data values with the associated longitudes and latitudes. There were regions where data are sparse, and it was assumed that soil conditions are more similar when two regions are closer to each other. The KDE was plotted with a color scale to represent locations and observed values of soil data. For clear visual presentations, the plot concentrated the areas of soil data collection by using the top 25% of estimated density values[48].

The location of each orchard can be identified by its longitude and latitude, and this information was used for regional presentations of quality over time. The average fruit size and sugar content at each orchard were averaged for the following three time periods: the fourth and fifth weeks of October, the first and second weeks of November, and the third and fourth weeks of November. The temporal averages were plotted at the given longitudes and latitudes (Fig. 3).

The observed trend of fruit size and sugar content with respect to each soil- and weather-related variable were visualized using smooth splines (Fig. 4). When the soil-related data were not available for the exact location of an orchard, the soil data were sorted according to the Euclidean distance from the orchard, and the averaged values of close locations were used as approximations. The regional-scale analysis was for descriptive purposes, and we state here that it was not for causal inference as the merged data were not obtained from an experiment.

## Inter-orchard analyses

Farmers' management practices are factors which distinguish between orchards. These are important as farmers can decide and control unlike weather conditions in open fields. Farmers self-reported their agricultural practices including thinning, mulching, spraying (pest control), fertilization, and pruning recorded from January to October in 2021. The frequency and time of each practice at each orchard varied, and the monthly frequency of each type of agricultural practice was visualized by ridge plot (Supplementary Fig. 1).

For demonstration purposes, we randomly selected two orchards (which grew the same cultivar) and compared their observed soil conditions, agricultural practices, and fruit quality (Fig. 5). We then separated the fruit quality data by orchard (27 orchards), ordered the orchards by the median sugar content and median fruit size, and used boxplots to graphically describe the inter-orchard variability of sugar content and fruit size (Fig. 6). In order to visualize the longitudinal patterns of mean sugar content and mean fruit size by orchard, we used the spaghetti plots between mid-October and late November (Supplementary Fig. 2).

To quantify the variation of sugar content and fruit size explained by orchard and harvest time, the $R^2$ was calculated under the mixed-effect model. This statistical model accounts for similarity of fruit quality within orchards, and it is useful particularly when observed fruit quality values within orchards are correlated which is a reasonable assumption as shown in Fig. 6 and Supplementary Fig. 2. The 'lme4'[49], 'lmerTest'[50], and 'MuMIn'[51] packages were used in R for this analysis. In this mixed-effect model, the orchard was treated as the random-effect, and the time was treated as the fixed-effect. Two-sided p-values were calculated for the relationship between time and fruit quality and the p-values are adjusted for the multiple testing[52].

An agricultural DT can be more valuable when orchard-level variables can predict the fruit quality. In addition to the aforementioned orchard-level variables, the orchard-level weather variables (temperature, humidity, and air pressure) were considered for predictive analysis of sugar content and fruit size. To gauge the predictive power of these orchard-level variables, an automatic machine learning (AutoML) algorithm was implemented with the following predictors: time (week and month), the three weather variables (temperature, humidity, and air pressure), the frequency of each of the five agricultural practices (thinning, mulching, spraying, fertilization, and pruning), and the orchard index. The AutoML algorithm compared the predictive performance of multiple machine learning algorithms and automatically selected the best one. The AutoML was implemented using the 'h2o' package[30] in R. The predicted values and observed values were plotted (Fig. 7), and the $R^2$, RMSE, and MAE were calculated to measure the predictive performance.

## Intra-orchard analyses

As aforementioned, one hundred trees were randomly selected and observed weekly in each orchard, and each tree was identified by a unique tag number. There were three levels of fruit position (low, middle, and high) determined based on the height from ground level. One fruit was taken from each position per tree per week, so there were a total of 300 fruit observed per orchard per week. One orchard was selected, and the hierarchical clustering analysis was used to group similar longitudinal trends for demonstration purposes (Fig. 8).

In addition to the orchard-level variation (inter-orchard), we assumed that the tree-level variation (intra-orchard) is another level of random-effect. We added the tree-level random-effect to the aforementioned mixed-effects model. Under this statistical model, orchard and tree were treated as the independent random-effects, the time and fruit position were treated as the fixed-effects. Two-sided p-values were calculated for the fixed-effects and adjusted for the multiple testing[52]. The predictability of the inter-orchard analysis and the intra-orchard analysis were compared by plotting the predicted values and

observed values of sugar content were visualized (Fig. 9). This comparison is to demonstrate potential benefit and necessity of tree-level analysis, in addition to the orchard-level analysis, for explaining the unknown source of variance in the fruit quality.

## Agricultural DT interface applet

Data collection and analysis are not sufficient for successful individualized agriculture, and farmers shall monitor individual trees on a regular basis. In this regard, it will be convenient if a user-friendly applet provides summaries of one's orchard (and comparison with other orchards in near locations) and a specific tree (and comparison with other trees in the orchard). For demonstrations with currently available data, we created an interface applet which is freely and easily accessible. If a user selects an orchard and then a tree of interest, the interactive applet provides the following information: (1) geographic location of the orchard, (2) regional soil components with RDA recommendations and comparison to other orchards in Jeju Island, (3) weather information with comparison to the other orchards, (4) agricultural practices with comparison to the other orchards, (5) inter- and intra-orchard comparison of sugar content, (6) inter- and intra-orchard comparison of fruit size, (7) inter- and intra-orchard history of sugar content, and (8) inter- and intra-orchard history of fruit size.

## Reporting summary

Further information on research design is available in the Nature Portfolio Reporting Summary linked to this article.

## Data availability

The data for the figures are provided in the Source Data file. All data (soil chemical properties, fruit quality, weather, agricultural practice, and the GIS map) used in this study have been deposited in the Github (https://github.com/heoseong/Digital_twin) and Zenodo repository (https://zenodo.org/records/10531851, https://doi.org/10.5281/zenodo.10531851)[53]. The data (soil chemical properties, fruit quality, weather, agricultural practice) used in this study are available in the open data portal (https:///www.data.go.kr) via Open API after registration and authorization process. The data on the GIS map of Jeju Island used in this study are available in the National Spatial Data Infrastructure Portal (https://www.vworld.kr) after registration and authorization process, and accessibility is limited to users in Korea. The regional codes of Jeju Island are available in the Ministry of the Interior and Safety (https://www.code.go.kr/stdcode/regCodeL.do), and the user must specify Jeju Island in Korean (제주특별자치도) in the dropdown menu of city/province (시/도) of the area selection (지역선택) to obtain the relevant data. Source data are provided with this paper.

## Code availability

All codes and associated data are available in the Github (https://github.com/heoseong/Digital_twin) and Zenodo repository (https://zenodo.org/records/10531851, https://doi.org/10.5281/zenodo.10531851)[53]. The source code of the applet (Shiny DT) is also available in the same Github and Zenodo repository.

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

## Acknowledgements
We thank Professor Yong Suk Chung (Jeju National University) who provided valuable expertise and input for this research and paper. We also thank the MOIS, RDA, JDC, NSDIP, and Kakao Developers in the Republic of Korea for creating and managing the data. There was no additional financial support for the data collection, data analysis, and the development of an agricultural DT interface applet demonstrated in this article.

## Author contributions
Conceptualization: S.H.; Data collection: S.H.; Methodology: S.K. and S.H.; Formal analysis: S.K. and S.H.; Figure preparation: S.K. and S.H.; Writing-original draft: S.H.; Writing-reviewing and editing: S.K. and S.H.; Supervision: S.H.; Website construction: S.K.

## Competing interests
The authors declare no competing interests.
