## [Peer Review File · Nature Communications]

Reviewers' Comments:

Reviewer #2:

Remarks to the Author:

62: "Total integration" is rather ambiguous? Regarding geospatial data, I have seen at least one paper which utilised such information in a digital twin for rural planning.

64: „all types" is rather ambiguous

70: ... in citrus production. Or the use of this approach with geospatial data. Explicitly outline the novel contribution/context.

Otherwise, a paper already exists that shows how decision making support can be achieved through data integration.

74: It is unclear if this refers to an publicly accessible api or an api that conforms to the open api specification?

74: Please reference the ministry or elaborate which country

105: Add a graphic or table explaining the difference between the outlined resolutions of analysis

137: Might be worthwhile to add a table outlining the key characteristics of these two orchards.

226: Ambiguous, if this is related to the digital twin then it's the first mention of a website. It should be explained how the digital twin will support decision making (I.e., how does the data flow from virtual-physical

The paper discusses many of the benefits applying digital twin might bring and the forms they could take, but lacks detail on open questions, design requirements, enabling components or methodologies that should be considered or investigated.

What is really missing is an explanation/understanding of why hasn't IA been achieved in any area of agriculture?

Reviewer #3:

Remarks to the Author:

This topic is very interesting and I think a lot of valuable research has been done by the researchers. However, I am not very convinced by the way the article currently is written/structured.

The introduction isn't captivating enough to my liking. The authors make a claim which I find tentative at best. They do not state why citrus orchards are special, what the challenges are, and how DTs can add value in that respect. The answers to that question gradually reveal themselves while I read through the subsequent sections, but I would have liked to know already early on in the article where this was going.

They do not highlight enough their real contribution to the field by developing this DT (because I read a lot of very interesting material later in the article that were not in the introduction). The data section starts with a (in my view) less interesting technical description of how data were retrieved from a portal. Yes, that open data portal is nice, but the intra- and inter-orchard and machine learning analyses that are described subsequently are much more interesting, in my opinion.

The article does not follow the traditional structure of introduction, methods, results, discussion,

conclusion. The various sections following the data section each present all at once the method description, the results, and discussion of those results. This a choice - I would have made a different choice.

The concluding remarks are well-written and make a convincing case about the use and added value of DT in orchards (not especially citrus, by the way).

Important messages of the article are sometimes too much hidden - like in a caption under Figure 1, or as an afterthought sentence at the end of a paragraph (e.g. line 201). The article could benefit from better presenting and highlighting its key messages. (And downtuning a bit what they now present as their key message).

The resolution of Figure 1 is a problem. Also I don't see why this figure is specific for citrus orchards. It looks like it is very generic (applicable to any agricultural crop). The title of the figure indicates this, but the caption says 'for citrus management'. And the caption of Figure 1 is outrageous - lots of information in there that don't belong in a caption.

I don't like much of the beginning of the article.

I like a lot of the remainder of the article.

The two parts should be better balanced - then this can become be a very interesting read.

More detailed comments: see in the attached pdf with my remarks in them.

Reviewer #4:

Remarks to the Author:

Dear authors

Thank you for submitting the article titled "Can digital twin technology make individualized-agriculture a reality?". My comments are as follows:

The article is well-structured and well-written. It brings an important insight into the use of digital twins towards individualizing agriculture. I believe there are some points to be clarified and improved. The first is related to the algorithms and statistical techniques used. Several of these techniques, including machine learning, were mentioned, but implementation details were not presented. The results of these algorithms have been little discussed from a quantitative point of view. Few numerical results of the machine learning algorithm were presented in the text. This makes the work non-reproducible.

Second, taking into account the implementation of the digital twin, it was not clear how the results of the algorithmic decisions in the plantation feed back into the virtual model since it must operate in a closed loop.

REVIEWER COMMENTS

We greatly appreciate helpful comments by the reviewers. We addressed every single item and responded below. Please note that we went through a major revision to address all reviewers' concerns and comments simultaneously.

Responses to Reviewer #1

1. (Line 62 in old version) "Total integration" is rather ambiguous? Regarding geospatial data, I have seen at least one paper which utilised such information in a digital twin for rural planning.

Response: After the major revision, the ambiguous language (total integration) is removed in the revised manuscript, and we tried to avoid any ambiguous expression throughout the paper.

2. (Line 64) „all types“ is rather ambiguous.

Response: This sentence, along with the above sentence (Line 62 in old version), has been removed and replaced with the following sentence: This article illustrates that through the integration of multiple datasets obtained from diverse sources (utilizing Open APIs) and the creation of a DT for mandarin orchard management, we can achieve not only precision agriculture (PA) but also individualized agriculture (IA), where each fruit tree is managed on an individual basis. [Line 119-123 in new version]

3. (Line 70) ... in citrus production. Or the use of this approach with geospatial data. Explicitly outline the novel contribution/context. Otherwise, a paper already exists that shows how decision making support can be achieved through data integration.

Response: As with the reviewer's comments, we read and cited papers that incorporated spatial data in detail. We have also demonstrated that our digital twin can support decision-making by integrating spatial data with fruit quality, agricultural practice, weather, and soil chemical properties data. Further, it was suggested that individualized agriculture, which can provide customized management for each tree, is possible. To demonstrate the possibility, we created a digital platform (<https://stevenkimcsumb.shinyapps.io/ShinyDT/>) which provides customized information tailored to the perspectives of stakeholders including policymakers, researchers, distributors, and farmers. We explicitly outlined this in the last paragraph of the introduction [Line 116-130].

4. (Line 74) It is unclear if this refers to an publicly accessible api or an api that conforms to the open api specification?

Response: We apologize for causing confusion to reviewers due to the ambiguous sentence. Once again, thank you for giving us the opportunity to improve this manuscript. We added details about the data resource and collection in the Methods section, and we clarified the unclear sentence as follows: Publicly accessible Open APIs, which are potential sources for developing an agricultural DT for mandarin orchard management, were collected from multiple sources on the data portal (<https://www.data.go.kr>). [Line 477-479]

5. (Line 74) Please reference the ministry or elaborate which country.

Response: We described the relevant details in the Methods section: "Data resources and collection through Open API. [Starting in Line 476]

6. (Line 105) Add a graphic or table explaining the difference between the outlined resolutions of analysis.

Response: For the regional scale analysis, we created Figs. 2, 3, and 4, according to your suggestion.

7. (Line 137) Might be worthwhile to add a table outlining the key characteristics of these two orchards.

Response: The key characteristics of the two orchards are graphically demonstrated in Fig. 6, and we added a table (Table 1) in the revised manuscript according to your suggestion. In addition, we described the spatial differences in soil characteristics between Hab and Iab orchards in the Inter-orchard analyses subsection of the Results section [Line 218-236]. The location, weather, agricultural practice, and fruit quality information of the Hab and Iab orchards can be also found on the following websites: <https://stevenkimcsumb.shinyapps.io/ShinyDT/>. In addition, we added the following sentences in the Results section:

- *Regional scale analyses section: According to the JARES report, the western part of Jeju Island is a non-volcanic ash soil area, which is similar to the land soil of the Korean Peninsula and is highly productive. On the other hand, the eastern part is a volcanic ash soil which is characterized by low OM and Av. P₂O₄ content and high Exch. Ca content. [Line 143-146]*
- *Inter-orchard analyses section: Hab in the western region and Iab in the eastern region, which grew the same cultivar (Miyagawa Wase). [Line 219-220]*
- *Inter-orchard analyses section: The Iab orchard had a low level of Av. P₂O₄ content and a high level of Exch. Ca content due to the characteristics of volcanic ash soil found in the eastern region. [Line 223-225]*

8. (Line 226) Ambiguous, if this is related to the digital twin then it's the first mention of a website. It should be explained how the digital twin will support decision making (I.e., how does the data flow from virtual-physical).

Response: We have revised Fig. 1 to demonstrate the process of data collection, data analysis, and decision-making support via the digital twin (DT). In addition, we created a website (<https://stevenkimcsumb.shinyapps.io/ShinyDT/>) to provide a concrete idea of the DT. Though our data were not complete due to the unexpected change in smart farm research policy and funding [Line 92-115], we were able to demonstrate inter-orchard and intra-orchard analysis. It is important for farmers to monitor individual trees when individualized agriculture is implemented, and the DT enables keeping track of fruit quality using the historical data and comparing an individual tree to other trees within the orchard. In addition, we elaborated the following in the subsection of intra-orchard analyses:

- *For instance, after identifying a group of trees producing fruits with low sugar content through hierarchical clustering analysis, customized agricultural practices can be applied to each selected tree to improve its sugar content. [Line 306-308]*
- *Therefore, trees with low sugar content may benefit from tailored agricultural practices such as rain-shelter cultivation, irrigation control, foliar fertilization, proper pruning, and thinning to enhance their sugar content. [Line 312-315]*

In the Methods section [starting in Line 474], we explained how the data from multiple sources are merged and analyzed, how the analyzed data flows through the pipeline to the DT, and how agricultural practices can be performed based on it. However, the current DT does not demonstrate how to automatically suggest agricultural practices in lacking areas and how to assess the effect of the agricultural practices, and these are beyond the scope of our paper and have not yet been implemented in the real world. To this end, it is not described in this paper, but it is briefly noted as limitations [Line 466-468].

9. The paper discusses many of the benefits applying digital twin might bring and the forms they could take,

but lacks detail on open questions, design requirements, enabling components or methodologies that should be considered or investigated.

Response: This comment is broadly addressed by the new structure of the revised paper, and we have described the results, discussion, and method of this DT in detail this time. More specifically we added the following paragraph in the Discussion section:

- *Improving or maintaining high fruit quality is both science and art, and farmers shall balance between empirical evidence and farmers' observations, experiences, and knowledge. At this point, it is an open question whether an individualized tree-level management will be more profitable than regional or orchard-level management. We need to consider how to lower the cost of implementing DT. The magnitude of benefits from implementing DT is unknown as of now, and we need a scientific approach to this question. We need to compare current regional or orchard-level practice versus new individualized agriculture guided by DT using a controlled randomized experiment. Dividing orchards into the control zone and experimental zone, it is necessary to confirm and estimate the benefit of DT. This study is limited to mandarin fruit with observational data, but we want to observe and experiment with more kinds of fruit. Unlike the current speed of technological advances, it will be a patient process. [Starting in Line 431]*

10. What is really missing is an explanation/understanding of why hasn't IA been achieved in any area of agriculture?

Response: This is a highly complex subject. We carefully address your question as follows:

- *For cereal or vegetable crops, when inbred lines are crossed, a uniform phenotype is produced in the F_1 offspring for specific traits due to the genetic uniformity. However, in reality, all these offspring are never truly identical and have only a narrow range of genetic variation. Moreover, these crops do not require extensive space to grow, making it even more difficult to investigate differences among individuals. Therefore, applying individualized agriculture (IA) to cereal or vegetable crops is exceedingly challenging. In contrast, fruit crops are relatively easy to study and apply IA because each tree requires ample space. Although fruit crops are propagated through asexual reproduction and have identical genomes, these identical clones exhibit significant variation depending on their environment. Thus, when researching orchards that cultivate a single cultivar (genotype, G), the variations in the phenotypes they exhibit are influenced by the environment (E) and agricultural practices (management, M). Researching this $G \times E \times M$ interaction remains highly challenging, and more experimentations are needed to address this complex question. From the consumers' perspective, regardless of the scientific merit of individualized agriculture, most consumers would not purchase a \$10 high-quality mandarin in South Korea and elsewhere. Future studies should address lowering the cost and labor in data collection, precision agriculture, and individualized agriculture. [Starting in Line 442]*

Responses to Reviewer #2

1. This topic is very interesting and I think a lot of valuable research has been done by the researchers. However, I am not very convinced by the way the article currently is written/structured. The introduction isn't captivating enough to my liking. The authors make a claim which I find tentative at best.

Response: We appreciate your review and comments. Regrettably, we initially prepared this paper in the format of Perspective, and it is now in the format of a research article following the suggestion by the editor and reviewers. We took this opportunity to improve our manuscript according to your comments. Consequently, we have added and enhanced the content in all sections (Introduction, Results, Discussion, and Methods). In particular, our claim and limitations are extensively discussed in the Discussion section.

2. They do not state why citrus orchards are special, what the challenges are, and how DTs can add value in that respect. The answers to that question gradually reveal themselves while I read through the subsequent sections, but I would have liked to know already early on in the article where this was going.

Response: The primary reason for selecting the mandarin orchard (related to Comment #10 of Reviewer #2) is that fruit crops were propagated through asexual propagation (grafting), making all trees genetically identical. Furthermore, unlike other crops, fruit crops require ample space per individual, which facilitates the collection of individual-specific data. Additionally, as fruit crops are perennial, information can be updated annually from the same individuals, enabling spatiotemporal analysis. We added these sentences in the Introduction section. [Line 87-91]

In accordance with other reviewers' suggestions, we have provided a new description in the Introduction section, explaining (1) why we considered mandarin (Citrus unshiu) orchards [Line 87-91; 117-118], (2) our current situation of the open-field smart farm research [Line 92-95], and (3) the value that this DT can create [Line 122-130]. The challenges (for any crops) are also discussed in the Discussion section [Line 442-458]. We are once again thankful for the opportunity to improve this manuscript.

3. They do not highlight enough their real contribution to the field by developing this DT (because I read a lot of very interesting material later in the article that were not in the introduction).

Response: We agree that we should have highlighted the real contribution early in the introduction. We substantially revised the last paragraph of the introduction section to highlight your point. [Line 116-130]

4. The data section starts with a (in my view) less interesting technical description of how data were retrieved from a portal.

Response: In response to the reviewer's comments, we removed all sentences describing our technical approach and moved the description of the data to the Method section. [Line 476-515] Given the current format in the revised manuscript, readers can choose to read or skip. Thank you.

5. Yes, that open data portal is nice, but the intra- and inter-orchard and machine learning analyses that are described subsequently are much more interesting, in my opinion.

Response: We added a description of statistical analysis and machine learning to the Method section. [Line 556-593]

6. The article does not follow the traditional structure of introduction, methods, results, discussion, conclusion. The various sections following the data section each present all at once the method description, the results, and discussion of those results. This is a choice - I would have made a different choice.

Response: As suggested by the editor and reviewers, we chose to follow the journal's format of a research article (Introduction, Results, Discussion, Methods).

7. The concluding remarks are well-written and make a convincing case about the use and added value of DT in orchards (not especially citrus, by the way).

Response: We rewrote the concluding remark at the end of the Discussion section. [Line 459-472] We made sure to explain the value of DT in orchards, and we carefully chose our language not to be specific to citrus only.

8. Important messages of the article are sometimes too much hidden - like in a caption under Figure 1, or as an afterthought sentence at the end of a paragraph (e.g. line 201).

Response: Following your suggestion, we reorganized the overall structure of this manuscript, and it helped us to deliver important messages more explicitly. We divided the discussion section into soil, weather, fruit quality, and agricultural practice DT and described its value, and explained the future of agricultural DT at the end of the discussion.

9. The article could benefit from better presenting and highlighting its key messages. (And down tuning a bit what they now present as their key message).

Response: We believe that this comment is related to your previous comment (#8). As you pointed earlier, some key messages are highlighted early in the paper (the last paragraph of the Introduction section), and we revealed the interactive applet (demonstration version: <https://stevenkimcsumb.shinyapps.io/ShinyDT/>) at the end of the introduction. We organized our key messages in the last subsection (Digital twin for fruit quality management) of the Discussion section including the practical challenges, open questions, and potential benefits of current and future DT. [Line 431-472]

10. The resolution of Figure 1 is a problem. Also I don't see why this figure is specific for citrus orchards. It looks like it is very generic (applicable to any agricultural crop).

Response: We improved the resolution of Fig. 1. There is a limitation by copying an image to the Word document, and we will consider further improvement (if needed) by submitting the source file if the manuscript is accepted. In addition, we removed "citrus" or "mandarin" in the figure and caption to make it more generic.

11. The title of the figure indicates this, but the caption says 'for citrus management'. And the caption of Figure 1 is outrageous - lots of information in there that don't belong in a caption.

Response: We shortened the caption of Fig. 1 and removed "citrus" or "mandarin" in the caption.

12. I don't like much of the beginning of the article. I like a lot of the remainder of the article. The two parts should be better balanced - then this can become be a very interesting read.

Response: As aforementioned, we substantially restructured our paper format. In particular, we revised the Introduction section to the best of our ability by including. We added more literature related to the DT and its implementation, briefly explained the smart farm research policy and publicly open sources available in South Korea, some key messages, and availability of our agricultural DT demo (interactive applet, R Shiny). We sincerely appreciate your critical comment.

13. More detailed comments: see in the attached pdf with my remarks in them.

13-1. (Old version_Line 15) So the aim is technological, 'let's see if we can combine all this into a DT'. That is

not the most interesting aim. I would really prefer to see a research question as the first aim, for which it is then needed to develop the DT. The audience would be more interested in how DT can solve actual problems. Although a digital twin is only a 'representation' of the object system, still we need a purpose or insight in the type of problem/question that can be addressed by the DT. I do not see that here.

Response: This first comment on the PDF document motivated the authors to carefully revise the abstract. Our goal was not to attempt technological integration, but it was a process to develop an agricultural DT and discuss what the DT could offer to stakeholders and the prospects for new agricultural systems in the future.

13-2. (Line 17) This is far too vague - your analyses show you did some very specific analyses and tests with your DT for much more concrete purposes. The later part of your article gives so much valuable information about the use and applicability of DT - why not also mention more of that in the abstract.

Response: The sentence is deleted while we rewrote the Abstract. [Line 10-26]

13-3. (Line 45) A bit weird to phrase it like this. There is always research going on, isn't there?

Response: Yes, it was. We removed the sentence and started the paragraph as follows:

- *There are a number of studies on agricultural digitalization using the above advanced technologies. [Line 57-58]*

13-4. (Line 46) Again a very technical approach: I would expect that it depends foremost on the aim of the DT, not the method or the data. If the angle is this technical (method-oriented), then also a more technical journal would be more suitable.

Response: We appreciate your point, and we agree. We deleted this sentence. In the second paragraph of the Introduction section, we provided literature related to DTs which used the advanced technologies mentioned in the first paragraph of the Introduction section.

13-5. (Line 58) This is a conference article that is over 6 years old, when DT hardly existed - it's not really a key example I would say.

Response: According to the reviewer's recommendation, this literature has been removed and we cited the latest journal articles.

13-6. (Line 62) This is a pretty bold statement and also I am puzzled by its meaning. "Only one data source"? I doubt it, more than one is most often the case. "Total data integration" - what is 'total'? "Including geospatial information"- several DT applications exist that include geospatial information, for example the work by Pylianidis. 'Lack of data set availability' - whose lack is that? The researcher's? The end user? Not clear to me.

Response: We agree that this sentence was somewhat provocative and misleading. We have removed this sentence.

13-7. (Line 68) This is nice, but not new in itself. If the only 'new' addition is that it is for a mandarin orchard this time, then that is not very convincing. Then I should first need to understand why mandarin/citrus is so much different from other crops.

Response: We agree with your comment, and removed this sentence. Instead, we revised the last part of Introduction. [Line 116-130]

13-8. (Line 73) "on the data portal" - which data portal? And why is this the first sentence of this section? - I

need to understand the 'what' and the 'why' first, before the 'how'. I see that there is a page long caption under Figure 1, outside this article, which explains the open data portal. I would appreciate if part of that text is transferred here, to help the reader understand it better.

Response: As suggested by reviewers, we moved the content of the caption to Introduction and Methods. [Line 95-115; 480-486]

13-9. (Line 77) Again: a lot of very technical details - why is this so interesting? (I miss the point)

Response: This part is deleted in the revised manuscript, and all technical details about data collection and parsing are included in the Methods section. [Line 477-515]

13-10. (Line 94) Obviously sugar content and fruit size are important. What else is relevant for citrus that makes this case so special? And are you interested in predicting yield? In improving quality? Just comparisons intra- and inter orchards?

Response: All you mentioned are very important and interesting to add. However, we do not have yield data. Yield data, as it is related to the individual benefit of farmers, cannot be included in public information. Unfortunately, we are keenly interested in this information, but it has not been disclosed. Therefore, we have focused on fruit quality (sugar content and fruit size) throughout the manuscript, and fruit yield is briefly described in the discussion. [Line 386-394]

Inter-orchard, intra-orchard, and regional scale summaries and analyses can provide customized information according to the interest of stakeholders (e.g., policymakers, researchers, farmers).

13-11. (Line 105) This is interesting and also the many variations are interesting/challenging.

Response: We agree with your view; interesting and challenging. In the major revision, we edited Fig. 2 and added Figs. 3 and 4 with explanations. [the subsection named Regional scale analyses; starting in Line 134]

13-12. (Line 119) How was this collected? By technological devices? By observation? By interviewing them? Self-reported through a portal? Otherwise?

Response: The data on agricultural practices were self-reported by the farmers, and they reported practice type, treatment amount, date, units, and agrochemical product name. [Line 490-492] This data was collected and managed by the JDC and released on the data portal, where we collected it using Open API. This is described in the Method section (the subsection named Data resources and collection through Open API; starting in Line 476).

13-13. (Line 135) I would recommend that that assumption (line 129-132) is the start of this paragraph, rather than the end of the previous paragraph.

Response: We deeply appreciate it. After careful consideration, we have deleted the relevant sentences (lines 129-132 in the old version), and the sentences after line 135 in the old version are stated after line 218 in the new version.

13-14. (Line 153) Also this section is impressive. This comparison of the fruit sizes and sugar content was carried out in the lab - that is manual data collection, not through open API, correct? I think this adds much value (in terms of validation and knowledge building) to the DT that wasn't mentioned at all in the first sections. (There the focus was only on the technicalities of combining various digital data sources)

Response: Yes, the measurement of fruit quality (sugar content and fruit size) is manual data collection. Once the data is uploaded, all the data is publicly available from JDC, and we can download the data. After this

point, we applied various statistical methods and software to demonstrate the DT. Following reviewers' suggestion, the technicalities of combining various digital data sources are moved to the Methods section. [Line 476-515] To avoid any confusion, we clearly stated that the fruit quality data were measured manually. [Line 496]

13-15. (Line 154) Am I correct in assuming that this was done manually?

Response: Yes. All of this data was measured manually by the JDC and is available via Open API from Data Portal in South Korea.

13-16. (Line 161) And now it's no longer manual.

Response: Yes, the downloaded data was analyzed by the authors using statistical software.

13-17. (Line 177) I would have liked to read more about citrus and how their quality is assessed in the introduction section

Response: We added the related sentences to the manuscript as suggested by the reviewer, but added it to the Inter-orchard analyses subsection of the Results to better fit the context of the article [line 211-218].

13-17. (Line 201) Please elaborate! This is the whole point of your article. What is that value? Those results? Were those insights new? Did we not know this before? How valid/reliable/useful do you deem the results?

Response: In the revised manuscript, we elaborated the point using appropriate statistics. When the five agricultural practices (inter-orchard level) and harvest time were considered, the model predicted sugar content with $R^2 = 0.19$; when the orchard index was given, the prediction improved as $R^2 = 0.38$; and when the tree index was given, the prediction improved as $R^2 = 0.66$. After showing these incremental results, we stated that the development of DT (which shows orchard- and tree-level information) will be valuable for individualized agriculture. When farmers apply some management practices at tree-level, they should monitor the fruit quality on a regular basis, and the DT makes the monitoring accurate and convenient. [Line 319-331]

13-18. (Line 213) Interesting and convincing. (Only do not claim that it hasn't been done before.)

Response: We sincerely apologize for the confusion caused by the incorrect sentence. We have modified this sentence as follows:

- *Using orchard-level soil chemical information from satellite imagery, it will be feasible to develop a DT which can generate regional and orchard-level maps of soil profile and predict fruit quality. [Line 348-350]*

13-19. (Line 225) For who? For the farmer? For the researcher?

Response: We used the language "stakeholders" to imply anyone who can use this information such as farmers, researchers, and policymakers. [Line 363]

13-20. (Line 236) I really like this. That's also why I would have liked to know more about how those management practices data were collected and integrated in the DT.

Response: Those management practice data were self-reported by farmers. [Line 490-492] The merged inter-orchard data allow us to compare the frequency of each management practice to all other orchards (among those who reported) in the DT. The interactive applet (the tab named "Agricultural Practice") reports the comparison for a selected orchard.

13-21. (Line 299) This section starts commenting on aspects that were not presented as this article's aim at all.

Response: We agree with your view, and we rewrote the section (Discussion) from scratch, and the concluding remark is at the end of the Discussion section.

13-21. (Line 299) Why is there a 2 page caption under Figure 1? Part of this would be very suitable in the text itself, for example the part on 'the open data portal'. I did not understand that when I read it in the text. I don't understand the function of all this text in regard to the rest of the article.

Response: We agree that it was a long caption. We trimmed the contents and added in the text. [Line 95-115]

13-22. (Fig. 1.) This sentence is the closest to an aim, interest, purpose of a DT that I have come across in this article. And it is written in the caption of a figure outside the article itself. Still: this is very generic and not very convincing to me. For yield prediction we need not only data (the focus of this article) but also models. The article explains

Response: We deleted the sentence as well as all sentences that mentioned yield prediction in the caption. We also admit that yield prediction requires models. We showed our results on fruit quality prediction in the results section. [Line 272-282; 319-331; Figs. 9 and 11]

13-23. (Fig. 4.) Figure 4: was>were

Response: We fixed it. [Fig. 6 in the revised manuscript] Thank you for your careful review.

13-24. (Fig. 4.) The scale of the y-axis of figure B suggests that we could also have half times - only integer values should be there

Response: We fixed the scale with integers only. [Fig. 6 in the revised manuscript] Thank you again.

Responses to Reviewer #3

1. The article is well-structured and well-written. It brings an important insight into the use of digital twins towards individualizing agriculture. I believe there are some points to be clarified and improved. The first is related to the algorithms and statistical techniques used. Several of these techniques, including machine learning, were mentioned, but implementation details were not presented. The results of these algorithms have been little discussed from a quantitative point of view. Few numerical results of the machine learning algorithm were presented in the text. This makes the work non-reproducible.

Response: We deeply appreciate your comments and suggestions. We would like to let you know that we initially prepared this paper in the format of Perspective. However, the editor suggested transitioning it into the format of a research article. Fortunately, we were granted the opportunity to thoroughly address reviewers' comments in this new format (with more words allowed). The revised manuscript now includes the Results and Methods section, and we added explanations of the statistical methods and AutoML in the Methods section. Regarding your point about the reproducibility, all data and codes have been made publicly available at GitHub (https://github.com/heoseong/Digital_twin) and Zenodo (<https://zenodo.org/records/8137834>). Finally, for readers' convenience, an interactive applet has been created in the revision process. (<https://stevenkimcsumb.shinyapps.io/ShinyDT/>). It demonstrates how the DT can support stakeholders such as farmers, researchers, distributors, and policymakers.

2. Second, taking into account the implementation of the digital twin, it was not clear how the results of the algorithmic decisions in the plantation feed back into the virtual model since it must operate in a closed loop.

Response: The authors concur with your opinion. Unfortunately, we were unable to delineate in this manuscript the tangible manifestation of actions in the real world following decision-making based on the information of DT, and how the outcomes of these actions are subsequently sensed and reflected back into the DT, as depicted by the blue line in the figure below.

Though our data were not complete due to the unexpected change in the smart farm research policy and funding, we combined all open data sources [Line 92-115], and we were able to demonstrate inter-orchard and intra-orchard analysis in the current form of DT. It is important for farmers to monitor individual trees when individualized agriculture is implemented, and the current DT enables keeping track of fruit quality using the historical data and comparing an individual tree to other trees within the orchard. Consequently, we added the potential and challenges of DT in the Discussion section [Line 466-472], acknowledging the necessity of these processes for the development of a comprehensive DT, albeit being unable to execute them within the scope of this study. Once again, we greatly appreciate your feedback which improved our manuscript.

Reviewers' Comments:

Reviewer #1:

Remarks to the Author:

The paper seems OK

Reviewer #2:

None